# Symbol-LLM: Leverage Language Models for Symbolic System in Visual Human Activity Reasoning

**Xiaoqian Wu**   **Yong-Lu Li**\*   **Jianhua Sun**   **Cewu Lu**\*

Shanghai Jiao Tong University

{enlighten, yonglu_li, gothic, lucewu}@sjtu.edu.cn

## Abstract

Human reasoning can be understood as a cooperation between the intuitive, associative "System-1" and the deliberative, logical "System-2". For existing System-1-like methods in visual activity understanding, it is crucial to integrate System-2 processing to improve explainability, generalization, and data efficiency. One possible path of activity reasoning is building a symbolic system composed of symbols and rules, where one rule connects multiple symbols, implying human knowledge and reasoning abilities. Previous methods have made progress, but are defective with limited symbols from handcraft and limited rules from visual-based annotations, failing to cover the complex patterns of activities and lacking compositional generalization. To overcome the defects, we propose a new symbolic system with two ideal important properties: broad-coverage symbols and rational rules. Collecting massive human knowledge via manual annotations is expensive to instantiate this symbolic system. Instead, we leverage the recent advancement of LLMs (Large Language Models) as an approximation of the two ideal properties, *i.e.*, Symbols from Large Language Models (**Symbol-LLM**). Then, given an image, visual contents from the images are extracted and checked as symbols and activity semantics are reasoned out based on rules via fuzzy logic calculation. Our method shows superiority in extensive activity understanding tasks. Code and data are available at `https://mvig-rhos.com/symbol_llm`.

## 1 Introduction

Human reasoning can be understood as a cooperation between two cognitive systems: the intuitive, associative "System-1" and the deliberative, logical "System-2" [25]. An example of System-2 reasoning is illustrated in Fig. 1a. A feather and an iron ball are dropped at the same height in a vacuum, which one will land first? Human intelligence derives the answer from the Universal Gravitation Law, *i.e.*, reasoning in a physical symbolic system to avoid intuitive errors. As a critical component in building AI systems, visual activity understanding urgently needs System-2 reasoning. Previous works [26, 17, 30, 19] typically establish the mapping between visual inputs and natural language, where knowledge is embedded into some non-symbolic form such as the weights of language models [24]. Though effective, these System-1-like methods depend on large-scale visual data, thus suffering from diminishing marginal utility. Also, it fails in explainability and generalization due to black-box computing. Thus, it is important to integrate System-2 reasoning.

With one glance at an image, we can effortlessly ground visual inputs to symbols/concepts and apply commonsense reasoning to imagine the world beyond the pixels, a process similar to the gravity case. Fig. 1b shows a possible path for System-2 visual reasoning for human activity: a symbolic system composed of symbols and rules. One rule "`Hip seated in a boat` ∧ `Hands control an engine` ∧ `Hands hold onto the side of a boat` ⊨ `Human ride a boat`" connects symbols

---

\*Corresponding authors.

37th Conference on Neural Information Processing Systems (NeurIPS 2023).

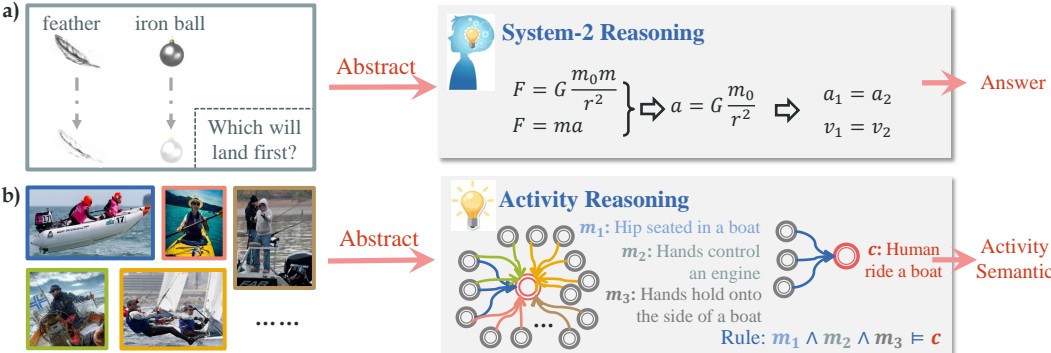

Figure 1: System-2 reasoning in **a)** solving physical questions and **b)** understanding human activities. They both involve a symbolic system implying human commonsense.

(*e.g.*, "Hands control an engine" "Human ride a boat") and implies human commonsense. Then, given an image, symbols are first extracted from visual inputs and then activity semantics are deduced based on rules. In this paper, we focus on this line of System-2 reasoning and zoom in from general visual understanding to human activities in various complex contexts, which is a challenging problem in visual understanding.

Current activity reasoning methods are not well equipped with System-2 reasoning. For instance, HAKE [21, 20] adopts a similar path as Fig. 1b from the perspective of human body part states and makes progress toward activity reasoning, but is defective with *limited symbols from handcraft* (93 primitives) and *limited rules* from visual-based annotations (100 K+ images). For example, Fig. 1b shows several rules related to the activity "Human ride a boat", *e.g.*,

(R1, green) Hip seated in a boat ∧ Hand hold onto the side of the boat ∧ Hand operate the steering wheel ⊨ Human ride a boat
(R2, orange) A rope is attached to a boat ∧ Hand hold the rope ∧ Hand adjust a sail ∧ Feet hold onto the rope ∧ Leg stretched out ⊨ Human ride a boat

However, the rules summarized by [20] are "(R3) Hip sit in ∧ Hand hold ⊨ Human ride a boat" "(R4) Hand hold ∧ Feet close with ⊨ Human ride a boat", which fail to cover the complex patterns of human activities. Thus, it lacks compositional generalization, *i.e.*, a generalized understanding of visual concepts, their relationships, and their novel combinations.

In view of this problem, we propose a novel symbolic system that implies abundant human knowledge and organizes diverse reasonable rules as in Fig. 1b. To achieve this, the symbols should be with **broad semantic coverage** to express various activity contexts. Moreover, the rules should be **rational and unambiguous**. R4 is a failure case, where 'Hand hold" and "Feet close with" do not necessarily lead to "Human ride a boat", and "Human tie a boat""Human exit a boat" are other possible conclusions. Ambiguous semantics should be corrected in rules to avoid confusion. Collecting massive human knowledge via manual annotation is expensive to instantiate this symbolic system. Our main insight is Symbols from Large Language Models (**Symbol-LLM**), *i.e.*, leveraging the recent advancement of Large Language Models (LLMs) to approximate the above two important properties. This is achieved by the *symbol-rule loop* and *entailment check* strategy. Then, the symbolic system can be used to reason with visual inputs, where visual contents from the images are *extracted and checked* as symbols and activity semantics are reasoned out via *fuzzy logic calculation*.

In conclusion, our main contributions are: **1)** We point out that current System-2 reasoning [20] suffers from a defective symbolic system with hand-crafted symbols and limited, ambiguous rules. **2)** To overcome the defects, we propose a novel symbolic system with broad-coverage symbols and rational rules. Further, we propose Symbol-LLM to instantiate it and show how it helps visual reasoning. **3)** In extensive activity understanding tasks, our method shows superiority in explainability, generalization, and data efficiency.

## 2 Related Work

**Activity Understanding** [22, 10] is still a difficult task compared with object detection [28] or pose estimation [9]. This is caused by its complex visual patterns and long-tail data distribution [32].

There are mainly image-based [5, 15], video-based [14, 4] and skeleton-based [31, 7] methods. In this paper, we focus on image-based methods.

**Vision-Language Models** (VLMs) have been intensively studied recently. Trained with web-scale image-text pairs, they can make zero-shot predictions on various recognition tasks [37] including human activities. Though effective, previous System-1-like methods [26, 17, 30, 19] need to be improved in explainability, generalization, and data efficiency. Thus, integrating System-2 reasoning, which is our main focus, is important.

**Probabilistic Programming** [27, 11] unifies probabilistic modeling and traditional programming. We follow the principles of the standard first-order logic (FoL) in probabilistic programming, *e.g.*, logical connectives $\wedge$, $\vee$. The standard FoL typically solves simplified relational tasks, where symbols are definitely True/False. However, when adapting FoL in vision reasoning tasks, the symbols are not known to be True/False. Instead, the models are required to predict the symbol probabilities (usually via a neural network). Thus, a hybrid "neuro-symbolic" method is adopted.

**Neuro-Symbolic Reasoning** provides a possible way of combining System-1 and System-2 learning, where knowledge is represented in symbolic form and learning and reasoning are computed by a neural network. [13]. Recently, researchers utilize neural-symbolic reasoning in tasks including visual question answering [2, 16, 34, 24, 29], decision-making [6], motion programming [18], scene understanding [23]. Focused on activity reasoning, HAKE [21, 20] propose an intermediate space spanned by human body part states as primitives. To overcome its inherent defect in compositional generalization, we propose a novel symbolic system with broad semantics and rational rules.

## 3 Method

### 3.1 Formulating the Symbolic System

#### 3.1.1 Structure

We first analyze the structure of the symbolic system. As aforementioned, the activity symbolic system is composed of rules that connect symbols, which implies human knowledge and reasoning abilities. Formally, one rule involves several symbols, where one of them is the *conclusion* and the others are *premises*. Each rule takes the form $r : m_1 \wedge m_2 \wedge \ldots \wedge m_{n_r} \models c$, where $M = \{m_i\}_{i=1}^{n_r}$ is the premise set and $c$ is the conclusion. Each conclusion corresponds to more than one rule with varied combinations of premise symbols because one activity typically has different visual patterns. Multiple rules for one conclusion are logically connected with $\vee$, *i.e.*, one conclusion holds if at least one rule is satisfied. A hyper-graph is a proper data structure to handle the complexity of the connection. Therefore, we define the structure of the activity symbolic system as a $\mathcal{B}$-Graph [12].

**Definition 1. (Directed Hypergraph [3])** *A hypergraph is a generalization of a graph in which an edge can join any number of vertices. A directed hypergraph is a pair $(\mathcal{X}, \mathcal{E})$, where $\mathcal{X}$ is a set of vertices, and $\mathcal{E}$ is a set of pairs of subsets of $\mathcal{X}$. Each of these pairs $(D, C) \in \mathcal{E}$ is a hyperedge; the vertex subset $D$ is its domain, and $C$ is its codomain.*

**Definition 2. ($\mathcal{B}$-Graph [12])** *A $\mathcal{B}$-graph is a type of directed hypergraph with only $\mathcal{B}$-arcs. A $\mathcal{B}$-arch is a type of a hyperedge that is directed to a single head vertex, and away from all its other vertices.*

**Definition 3. (Activity Symbolic System)** *The activity symbolic system $(\mathcal{S}, \mathcal{R})$ is a $\mathcal{B}$-Graph, where $\mathcal{S}$ is a set of vertices/symbols, and $\mathcal{R}$ is a set of pairs of subsets of $\mathcal{S}$. Each of these pairs $r = (M, C) \in \mathcal{R}$ is a hyperedge/rule; the vertex subset $M = \{m_i\}_{i=1}^{n_r}$ is its domain/premises, and $C = \{c\}$ is its codomain/conclusion. Equivalently, a rule takes the form $r : \bigwedge_{m \in M} m \models c$. Since $|C| = 1$, $r$ is a $\mathcal{B}$-arch.*

**Definition 4. (Decomposition of Activity Symbolic System)** *One conclusion $c^*$ corresponds to one symbolic sub-system $(\mathcal{S}_{c^*}, \mathcal{R}_{c^*})$, where $\mathcal{S}_{c^*} \subseteq \mathcal{S}$, $\mathcal{R}_{c^*} \subseteq \mathcal{R}$. $\forall r = (M, C) \in \mathcal{R}_{c^*}$, $C = \{c^*\}$. $\forall s \in \mathcal{S}_{c^*} \backslash \{c^*\}$, $\exists r = (M, C) \in \mathcal{R}_{c^*}$, $s \in M$.*

Def. 3,4 is depicted in Fig. 2. Def. 3 is based on Def. 1,2 and analysis above. In applications, we typically judge a specific conclusion with other symbols/rules removed. It is achieved by decomposing the activity symbolic system (graph) into sub-systems (sub-graphs) in Def. 4.

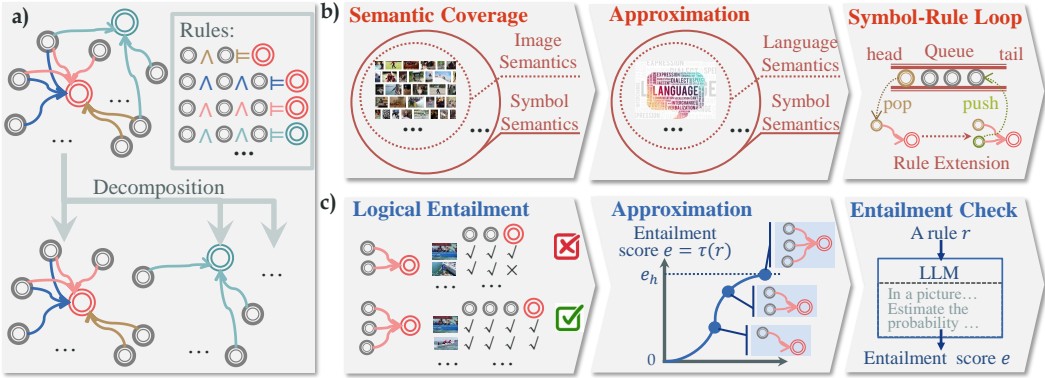

Figure 2: Our activity symbolic system. **a)** Structure and decomposition of the symbolic system (Sec. 3.1.1). **b)** Semantic coverage (Sec. 3.1.2). It can be approximated based on LLMs' knowledge and achieved via the symbol-rule loop (Sec. 3.2.1). **c)** Logical entailment (Sec. 3.1.2). It can be approximated based on an entailment scoring function and achieved by entailment check (Sec. 3.2.2).

#### 3.1.2 Two Ideal Properties

Existing methods [21, 20] can also be formulated under this structure. However, they fail to cover the complex patterns of activities and lack compositional generalization. To overcome the defects, we propose two important ideal properties: 1) symbols should be with broad *semantic coverage* to express different conditions in the activity image database; 2) rules should satisfy *logical entailment* [8] to add rationality and avoid ambiguity, *i.e.*, the premises set should lead to the conclusion without exception. They are formulated in Def. 5,6 and illustrated in Fig. 2b,c.

---

**Definition 5. (Semantic Coverage of Activity Symbolic System)** *Given a very large-scale activity images database* $\mathcal{D} = \{(I, A_I, \mathcal{S}_I)\}$ *(I: image, $A_I$: ground-truth activities happening in I, $\mathcal{S}_I$ : ground-truth symbols happening in I), then* $\forall (I, A_I, \mathcal{S}_I) \in \mathcal{D}, \forall s \in \mathcal{S}_I, s \in \mathcal{S}.$

**Definition 6. (Logical Entailment of Activity Symbolic System)** $\forall r = (M, C) \in \mathcal{R}$, $M = \{m_i\}_{i=1}^{n_r}$, *we have:*

1. $\forall (I, A_I, \mathcal{S}_I) \in \mathcal{D}$, *if* $M \subset \mathcal{S}_I$, *then* $c \in A_I$;
2. $\forall 1 \leq i \leq n_r, \exists (I, A_I, \mathcal{S}_I) \in \mathcal{D}, M \backslash \{m_i\} \subset \mathcal{S}_I$, *but* $c \notin A_I$.

---

### 3.2 Instantiating the Symbolic System

To instantiate the proposed symbolic system, it is expensive to collect massive human knowledge via manual annotation as in HAKE [21, 20]. Instead, we use LLMs which have profoundly reshaped the acquisition of human knowledge and shown impressive language reasoning capabilities, *i.e.*, Symbol-LLM framework. In Sec. 3.2.1 and Sec. 3.2.2, we discuss the approximation of the two ideal properties with the help of LLMs and provide practical solutions, then a summarized pipeline is presented in Sec. 3.2.3.

#### 3.2.1 Solving Semantic Coverage

**Approximation.** In Def. 5, the target domain of semantic coverage is a very large-scale activity image database, which is too costly to implement. For instance, HAKE [21] collect symbols from 100 K+ image via manual annotations, which is a heavy workload but still very limited. Here, the target domain can be replaced with semantic coverage of language models since natural language is another carrier of human commonsense knowledge. LLMs profit from large-scale text pre-training and imply abundant text-based human knowledge, thus can handle complex visual activity patterns without extra visual data. The approximation of Def. 5 is:

---

**Definition 7. (Approximation of Semantic Coverage of Activity Symbolic System)** *Given an LLM $\mathcal{L}$ and an activity set $\mathcal{A}$, $\forall A \in \mathcal{A}$, $\mathcal{L}$ implies a symbol set $\mathcal{S}_A$ as premises of A, then* $\forall s \in \mathcal{S}_A, s \in \mathcal{S}.$

---

**Symbol-Rule Loop.** In the implementation, generating rules connecting the symbols is non-trivial and challenging. Intuitively, we can ask LLMs "What is the rule of the activity". However, the instruction is ambiguous for LLMs to generate satisfying answers. More information should be provided to generate rules. Also, it is costly to generate all symbols and exhaustively query their connections. Thus, we propose to adopt *symbol-rule loops* to generate them alternately. Given a known symbol, we can add another symbol and extend it into a rule via prompting:

```
(Rule Extension)
In a picture, IF [<known symbols>] AND [condition] THEN [<activity>].
[condition] is one concise phrase.  The format is "<The person's
hands/arms/hip/legs/feet> <verb> <object>".  What is [condition]?
```

Then, the answer for "[condition]" is the extended symbol. Also, a candidate rule "<known symbols> $\wedge$ <extended symbol> $\models$ <activity>" is generated. Thus, we get a new extended symbol from this known symbol, and they are connected with rules. The new symbol can be repeatedly used as a known symbol to generate new rules. As is shown in Fig. 2b, a known symbol (the brown circle) is first "popped out" from a queue, and new symbols (the green circle) and rules (the pink edge) are generated by extending the known symbol. Then, the new symbol is "pushed in" to start the next loop. Thus, LLMs' implicit knowledge is fully exploited to satisfy Def. 7. To get the initial symbols, we ask an LLM to describe hand-related states related to the activity. The prompts mentioned above are determined via a trial-and-error process by human experts on randomly sampled activities (detailed in supplementary).

### 3.2.2 Solving Logical Entailment

**Approximation.** In Def. 6, it is costly to verify the logical entailment upon a very large-scale activity image database. Instead, we develop an entailment scoring function from an LLM based on its knowledge and language reasoning capability. As illustrated in Fig. 2c, as the rule extends from $m_1 \models c, m_1 \wedge m_2 \models c$ to $m_1 \wedge m_2 \wedge m_3 \models c$, new symbols are successively added, and the measured entailment score increases. With the scoring function, the approximation of Def. 6 is:

---

**Definition 8. (Approximation of Logical Entailment of Activity Symbolic System)** *Given a function $\tau(\cdot)$ to measure the entailment score of a rule, and an entailment threshold $e_h$, then $\forall r = (M, C) \in \mathcal{R}$, $M = \{m_i\}_{i=1}^{n_r}$, we have:*

1. *$\tau(\bigwedge_{m \in M} m \models c) \geq e_h$;*
2. *$\forall 1 \leq i \leq n_r, \tau(\bigwedge_{m \in M \setminus \{m_i\}} m \models c) < e_h$.*

---

**Entailment Check.** As is shown in Fig. 2c, to implement the scoring function based on an LLM, we rewrite the rule $r$ as a sentence and design the prompt as:

```
(Entailment Check)
In a picture, <symbol 1>, <symbol 2> ... <symbol n_r>.  Estimate the
probability that he is <activity> at the same time.  Choose from:  (a)
0.1, (b) 0.5, (c) 0.7, (d) 0.9, (e) 0.95, (f) unknown.
```

Then, the output answers can be used as the entailment score $e = \tau(r)$, whose credibility depends on the knowledge from the LLM. For one rule, the answering text is sampled five times and their average scores are taken as the final result to add stability. In practice, we set $e_h = 0.9$.

### 3.2.3 Summarized Pipeline

The pipeline of the symbolic system can be summarized as: given a target activity set $\mathcal{A}$, or equivalently, a conclusion set $\mathcal{C}$, we generate a sub-system $(\mathcal{S}_c, \mathcal{R}_c)$ for each $c \in \mathcal{C}$ (Def. 4). The symbolic system is merged as $(\mathcal{S}, \mathcal{R}) = (\bigcup_{c \in \mathcal{C}} \mathcal{S}_c, \bigcup_{c \in \mathcal{C}} \mathcal{R}_c)$. Alg. 1 shows how to get the sub-system $(\mathcal{S}_c, \mathcal{R}_c)$ for a given conclusion $c$. Details are explained in Alg. 1 comment and the supplementary.

### 3.3 Reasoning with Visual Inputs

With the activity symbol system constructed, we can utilize it to reason with visual inputs. As is shown in Fig. 3, visual information is extracted and checked based on the defined symbols and

**Algorithm 1** Instantiating the Symbolic System

**Input:** conclusion $c$, entailment threshold $e_h$
**Output:** Symbolic Sub-System $(\mathcal{S}_c, \mathcal{R}_c)$
1: $\mathcal{S}_c^0 \leftarrow$ Symbol_Initialization $(c)$
2: $\mathcal{S}_c^{cand}.push(\mathcal{S}_c^0)$                                      $\triangleright$ A queue $\mathcal{S}_c^{cand}$ stores symbols
3: $\mathcal{S}_c \leftarrow \{c\}, \mathcal{R}_c \leftarrow \{\}$
4: **while** not $\mathcal{S}_c^{cand}.is\_empty()$ **do**
5:     $m_{kno} \leftarrow \mathcal{S}_c^{cand}.pop()$            $\triangleright$ A known symbol $m_{kno}$ is taken for rule extension
6:     $M = \{m_{kno}\}$                $\triangleright$ The premises set $M$ is set for the current rule
7:     **while** Entailment_Check$(M) < e_h$ **do**
8:         $m_{new} \leftarrow$ Rule_Extension$(M, c)$
9:         $M = M \cup \{m_{new}\}$               $\triangleright$ A new symbol $m_{new}$ is added
10:     **end while**
11:     $\mathcal{S}_c \leftarrow \mathcal{S}_c \cup M$                 $\triangleright$ The symbol set $\mathcal{S}_c$ is updated
12:     $\mathcal{R}_c \leftarrow \mathcal{R}_c \cup \{r : \bigwedge_{m \in M} m \models c\}$    $\triangleright$ The rule set $\mathcal{R}_c$ is updated
13:     $\mathcal{S}_c^{cand}.push(M \backslash \mathcal{S}_c)$    $\triangleright$ $M$ is added to $\mathcal{S}_c^{cand}$ with redundant symbols removed
14: **end while**
15: **return** $(\mathcal{S}_c, \mathcal{R}_c)$

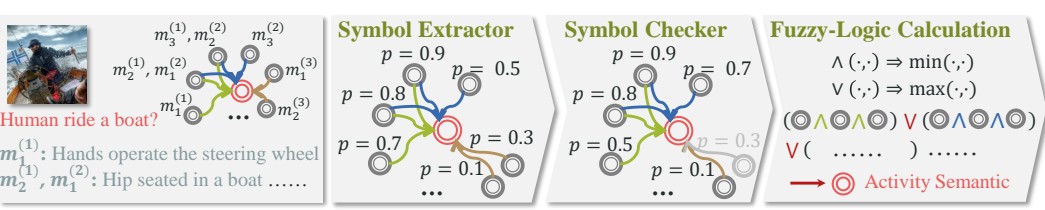

Figure 3: Visual reasoning with the proposed activity symbol system.

organized as a probability distribution on the hyper-graph. Then, the activity semantics can be reasoned out based on the rules and the probability of the symbols. We detail the key steps as follows.

**Decomposition.** Given the input image and the target activity/conclusion $c$, we first decompose the symbol system to exclude unrelated symbols and rules based on Def. 4. Thus, we get the rule set $\mathcal{R}_c = \{r^{(j)}\}_{j=1}^{N_c}$, where a rule takes the form $r^{(j)} : m_1^{(j)} \wedge m_2^{(j)} \wedge \ldots m_{n_{r^{(j)}}}^{(j)} \models c$.

**Extracting Visual Symbols.** Given an image, to extract visual content and express it in a symbolic form, we measure the probability of each defined symbol (Fig. 3). In [20], the visual extractor is a customized model trained with annotations, which is costly, especially for our symbolic system with broad-range symbols. Instead, we utilize a simplified visual extractor based on existing System-1-like VLMs. To determine the probability of a symbol, we convert it into a text question and query the answer from a VLM. For example, given a premise symbol $m_i^{(j)}$ "`Hip seated in a boat`", it is converted into a sentence "The person's hip is seated in a boat. Yes/No?". Used as a scoring function [1], the language model outputs the probability of answering "Yes", "No" as $p_{y,i}^{(j)}, p_{n,i}^{(j)}$. Then the probability $p_i^{(j)}$ of the symbol can be obtained via normalizing the Yes/No answer as

$$p_i^{(j)} = \frac{e^{p_{y,i}^{(j)}}}{e^{p_{y,i}^{(j)}} + e^{p_{n,i}^{(j)}}}. \tag{1}$$

**Checking Visual Symbols.** To ensure the reliability of the symbols' probabilities, we developed a checker module to filter out visual symbols with uncertain predictions. Each symbol is paraphrased into several variants, and the predictions of variants should be similar due to semantic consistency. Thus, the standard deviation of the predictions can be used to filter symbols with uncertainty.

**Reasoning.** With the symbol extractor and checker, the probability $p_i^{(j)}$ for each premise symbol $m_i^{(j)}$ in the rule set $\mathcal{R}_c$ is measured. Then, the probability $p_c$ of the conclusion is reasoned based on premise symbols' probabilities and the rules. Premise symbols within a rule are connected with $\wedge$, while rules for a conclusion are connected with $\vee$. We utilize the fuzzy-logic calculation [35], where

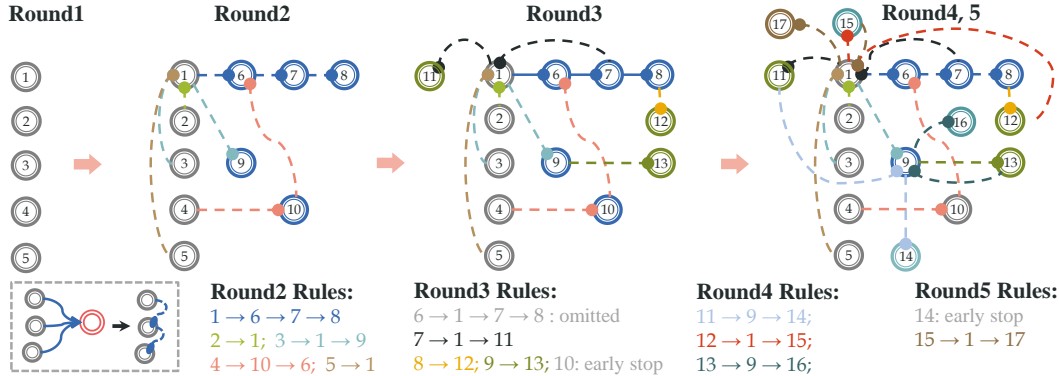

Figure 4: Detailed process for instantiating the symbolic sub-system of the activity "Human board an airplane". As is shown in the left bottom box, there is some difference in graph presentation between this figure and Fig. 1-3: here the edge connects premises instead of connecting premise and conclusion. The numbers 1-17 correspond to the 17 symbols listed in Tab. 1. The "→" in "1→6→7→8" means that 6, 7, 8 is obtained via 1's rule extension one after another.

| | Symbols | |
|---|---|---|
| | (1) hold a boarding pass | (2) place luggage in overhead compartment |
| **Round1** | (3) adjust seatbelt | (4) wave goodbye to loved ones |
| | (5) grip a luggage handle | |
| | (6) walk towards the boarding gate | (7) luggage visible beside him |
| **Round2** | (8) boarding pass is scanned by airport staff | (9) stand on the jet bridge |
| | (10) luggage is loaded onto the plane | |
| **Round3** | (11) reach for the airplane door handle | (12) stand in line with carry-on luggage |
| | (13) hold the carry-on luggage | |
| **Round4** | (14) open the airplane door | (15) move forward in the line |
| | (16) move towards the airplane door | |
| **Round5** | (17) airline staff checking the boarding pass | |

Table 1: Generated symbols & rules for "human board an airplane". Round $i$: $i$-th symbol-rule loop.

the operator $x \wedge y$, $x \vee y$ are replaced with $min(x,y)$, $max(x,y)$ respectively. Thus, we have

$$p_c = max_{1 \leq j \leq N_c}(min_{1 \leq i \leq n_{r(j)}} p_i^{(j)}). \tag{2}$$

The fuzzy-logic calculation is more explainable and lightweight than the logical modules in [20], whose parameters need extra training samples. Benefiting from the sound definition of our symbolic system, it has the potential to achieve *better* performance even with simplified reasoning calculations.

## 4 Experiment

### 4.1 Dataset and Metric

We conduct experiments on image-level activity understanding benchmarks with diverse tasks: **HICO** [5] (Human-Object Interaction (HOI) recognition), **Stanford40** [33] (action recognition), **HAKE** [21]-**Verb** (verb recognition), **HAKE** [21]-**PaSta** (conditional PaSta Q-A). HAKE [21]-Verb and HAKE [21]-PaSta is the newly constructed benchmark based on HAKE [21] data. We adopt mAP metric for HICO [5], HAKE [21]-Verb and Stanford40 [33], and top-1 accuracy metric for HAKE [21]-PaSta. For more details, please refer to the supplementary.

### 4.2 Symbolic System Experiment

**An example.** The LLM used is OpenAI GPT3.5. To demonstrate the instantiation of our symbolic system, we take the activity "human board an airplane" as an example. Tab. 1 shows the symbols and rules generated, and Fig. 4 illustrates the detailed process. In the 1st round, 5 initial symbols are generated. Then in the 2nd round, they are extended into rules satisfying entailment. Later, the new symbols are used in the next round. Finally, 17 symbols and 12 rules are generated within 5

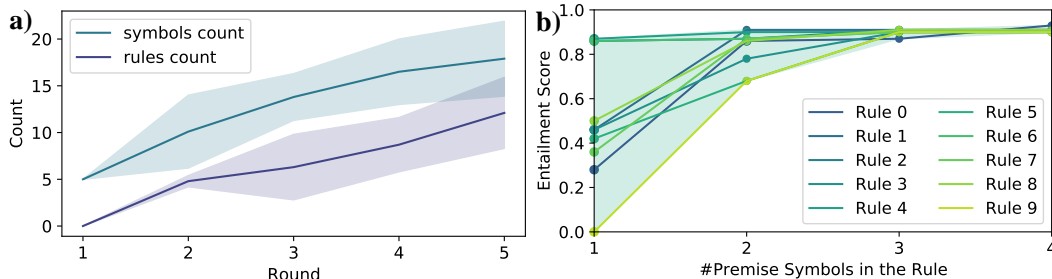

Figure 5: Statistics of the symbolic system. a) Accumulated symbols & rules count in each round. The data is from 50 randomly sampled activities. The average value and fluctuation range are shown. b) Increased entailment scores as the 10 randomly sampled rules extended with more premise symbols.

rounds. In practice, at most 15 symbols are used for rule extension to avoid semantic redundancy. For example, in Fig. 4, symbols 16 and 17 are not used for rule extension, leading to a stop in the 5th round. From Fig. 4 and Tab. 1, we can find that diverse symbols are mined, including corner cases such as "adjust seatbelt" "reach for the airplane door handle", verifying the effectiveness of the proposed symbolic system. Some special cases are shown as gray text. The rule "6→1→7→8" is equivalent to the existing rule "1→6→7→8", thus omitted. Extension from the symbols 10 and 14 stops early because adding premises nevertheless leads to a drop in entailment scores. In practice, multiple rules are extended from a known symbol to increase diversity.

**Entailment.** Fig. 5 shows statistics of the symbolic system. In Fig. 5a, we randomly sample 50 activities and show the accumulated symbols and rules count in each round, verifying the diversity of the generated symbols and rules. In Fig. 5b, we randomly sample 10 rules and show the entailment scores change as the rule extends with more premise symbols. For example, for rule 0: $m_1 \wedge m_2 \wedge m_3 \wedge m_4 \models c$, the entailment score for $m_1 \models c, m_1 \wedge m_2 \models c, m_1 \wedge m_2 \wedge m_3 \models c, m_1 \wedge m_2 \wedge m_3 \wedge m_4 \models c$ are 0.28, 0.86, 0.87, and 0.93, respectively. The score is calculated by querying 5 times and averaged. We find a climb up in the entailment score , implying the effectiveness of the entailment check. Once the entailment score $e$ surpasses the threshold $e_h = 0.9$, it is regarded as a rule equipped with logical entailment and updated into the symbolic system.

**Performance Evaluation.** To evaluate the symbolic system, we construct a SymAct (Symbol Activity) test set. It is a small subset of the HICO test set (120 images for 10 activity classes). For each image and each of its ground-truth activities, we list all of the symbols related to the activity in the symbolic system and annotate whether a symbol happens (0/1) in the image via human judgment. We find the proposed symbolic system has: **1) broader semantic coverage** than HAKE. For quantitative measurement, we count the happening symbols for each image-activity pair under different symbolic systems. The average number is 10.8 for ours and 1.8 for HAKE. **2) more rational rules** than HAKE. We evaluate confusion by counting different image-activity pairs which share the same symbols. For the HAKE symbolic system, the confusion problem is severe, with 261 confusion pairs over $C_{120}^2 = 7140$ pairs (accounting for 3.7%). For our symbolic system, there are no confusion pairs on the test set because of the presentation ability of symbols and entailment check.

**Robustness.** We analyze the robustness of the generated symbolic system. We find it satisfies: **1) Convergence.** After a certain stopping criterion, a new round will bring few or even no new symbols to the sub-system. In practice, the stopping criterion is: At most 15 symbols are used for rule extension to avoid semantic redundancy. After that, new symbols are not used for rule extension, leading to a stop of the loop. **2) Low sensitivity to initial conditions.** The symbol-rule loop provides a reliable and stable symbolic sub-system over a range of initial conditions. **3) Low sensitivity to different prompts.** Experiment results are less sensitive to prompts with minor differences, *i.e.*, slightly paraphrasing the prompts. For more details, please refer to the supplementary.

### 4.3 Visual Reasoning Experiment

**Implementation and Settings.** For visual reasoning, we use BLIP2 [19] ViT-g FlanT5-XL model to extract visual symbols. BLIP2 [19] is a visual-language pre-trained model with a frozen large language model, thus well equipped with question-answering abilities to effectively extract symbols.

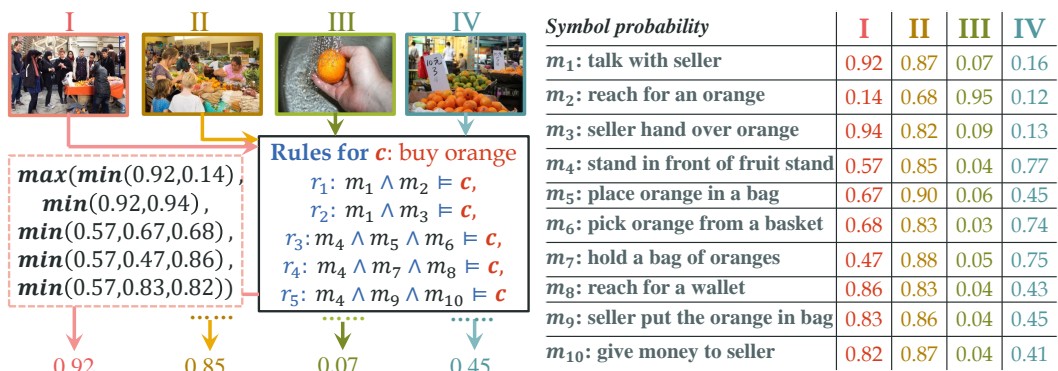

Figure 6: Visualization results. Symbols & activity predictions for "human buy orange" are shown.

| Method | HICO mAP | | HAKE-verb mAP | | Stanford-40 | HAKE-PaSta |
| | fine-tuned | zero-shot | fine-tuned | zero-shot | zero-shot mAP | zero-shot Acc(%) |
|---|---|---|---|---|---|---|
| CLIP [26] | 67.12 | 37.08 | 73.82 | 43.92 | 75.68 | 39.36 |
| CLIP [26]+Reasoning | **69.73** | **43.21** | **75.27** | **48.95** | **82.22** | **40.47** |
| BLIP2 [19] | - | 50.61 | - | 49.47 | 91.85 | 43.81 |
| BLIP2 [19]+Reasoning | - | **53.15** | - | **51.37** | **92.59** | **44.65** |

Table 2: Results on activity benchmarks. More HICO [5] baselines are listed in supplementary.

As a *plug-and-play*, the reasoning is compatible with existing System-1-like methods. We follow the setting of HAKE [20] to integrate the reasoning result with the prediction of System-1-like methods. We mainly choose 2 typical System-1-like methods: 1) CLIP [26]: a visual-language model pre-trained with large-scale image-text pairs and a contrastive objective, where we use its ViT-L/14 model with a resolution of 336 pixels; 2) BLIP2 [19]: with its image-to-text generation ability, it functions either as a System-1-like baseline, or as the symbol extractor for reasoning. To fully test the ability of the models, we adopt both "fine-tuned" and "zero-shot" settings, where the former model is fine-tuned on the training set of the dataset.

**Results.** The results of visual reasoning are shown in Tab. 2. Comparing the two baselines, we find that with a frozen language model, BLIP2 [19] is more capable of understanding activity semantics and outperforms CLIP [26] in various zero-shot benchmarks. Under the fine-tuned setting, our method achieves **69.73** and **75.27** mAP respectively on HICO [5] and HAKE [21]-Verb, which outperforms the baseline CLIP [26] with **2.61** and **1.45** mAP respectively, verifying its effectiveness. Under the zero-shot setting, our method also outperforms existing System-1-like methods. The improvement compared with CLIP [26] is: **6.13** mAP (HICO [5]), **5.03** mAP (HAKE [20]-Verb), **6.54** mAP (Stanford40 [33]), **1.11**% acc (HAKE [20]-PaSta). The improvement compared with BLIP2 [19] is: **2.54** mAP (HICO [5]), **1.90** mAP (HAKE [20]-Verb), **0.74** mAP (Stanford40 [33]), **0.84**% acc (HAKE [20]-PaSta). In conclusion, our method brings performance improvement and shows significant zero-shot *generalization* abilities in various activity understanding tasks. Notably, the improvement is achieved by exploiting knowledge instead of adding visual training samples. The exploited knowledge can be reused for future study. Thus, our method is superior in *data efficiency*. Further, we visualize the results in Fig. 6. The rules and the logic calculation add *explainability* to the inference process. Also, they help to correct false negative (I, II in Fig. 6) and false positive (III, IV) cases, thus improving compositional generalization.

## 4.4 Ablation Study

We conduct ablation studies on HAKE [20]-Verb zero-shot setting and list the results in Tab. 3.

**Reasoning.** We first verify the effectiveness of integrating System-2 reasoning. Results of CLIP [26] baseline can be either combined with reasoning results or trivially combined with other baseline results (CLIP [26]+BLIP2 [19]). We find that the former is superior (48.95 mAP) to the latter (44.76 mAP), though both outperform the baseline CLIP [26] (43.92 mAP). It verifies the necessity of System-2 reasoning other than trivially combining predictions from two models.

**Symbolic System.** We then verify the effects of the symbolic system on visual reasoning. We adopt the symbolic system in HAKE [20] and find a degradation from 48.95 to 44.98 mAP, as its defects in semantic coverage and rule rationality, and the short-age is amplified in the fuzzy logic calculation without learnable parameters. Next, we construct the symbolic system without logical entailment, where each rule has 3 premise symbols without checking its entailment score. It leads to a dropped performance because of rules with errors or incomplete semantics. Then, the symbol-rule loop is removed, with only rules in the 1st and 2nd rounds updated to the symbolic system. As the diversity of rules decreases, the model suffers from a decreased performance. The importance of diversity is further verified by the performance drop when only 80%, 50%, and 20% randomly sampled rules are used.

| Method | mAP |
|---|---|
| CLIP [26]+Reasoning | **48.95** |
| CLIP [26]+BLIP2 [19] | 44.76 |
| CLIP [26] | 43.92 |
| rules from HAKE [20] | 44.98 |
| w/o entailment check | 46.51 |
| w/o symbol-rule loop | 47.53 |
| 80% rules | 48.57 |
| 50% rules | 47.61 |
| 20% rules | 46.22 |
| CLIP [26] as extractor | 46.09 |
| w/o checking symbols | 48.17 |

Table 3: Ablation studies on zero-shot HAKE [20]-Verb.

**Symbols Extractor/Checker.** Next, we replace the symbol extractor BLIP2[19] with CLIP [26] and find a performance fall with a weaker ability to extract visual information. Also, reasoning without a symbol checker suffers from degradation due to the negative effect of inaccurate symbol predictions.

## 4.5 Bottleneck Analysis

Based on the SymAct test set, we make further analysis of the bottleneck of System-2 reasoning. We investigate the impact of symbol prediction and symbolic system separately. With perfect symbol predictions, we assume the probabilities of symbols are definitely known as 0/1 instead of $p \in [0,1]$. The ground-truth symbols are annotated as explained in Sec. 4.2. With a perfect symbolic system, we assume that the generated rules cover all samples (image-activity pairs).

The performance drop ($100.00 \rightarrow 60.79$ mAP, $83.11 \rightarrow 41.52$ mAP) indicates the symbolic system errors, *i.e.*, the generated rules have not covered all samples (image-activity pairs). The drop ($100.00 \rightarrow 83.11$ mAP, $60.79 \rightarrow 41.52$ mAP) indicates the errors of symbol predictions, *i.e.*, symbol probabilities cannot be predicted accurately. In the supplementary, we also provide two failure cases that emerge from symbol prediction errors and symbolic system errors respectively.

| Symbol Prediction | Symbolic System | mAP |
|---|---|---|
| perfect | perfect | **100.00** |
| imperfect | perfect | **83.11** |
| perfect | imperfect | **60.79** |
| imperfect | imperfect | **41.52** |

Table 4: System-1/2 analysis on SymAct test set.

## 5 Conclusion and Discussion

In this work, we rethink the symbolic system in activity reasoning and propose a new one with broad-coverage symbols and rational rules. Thus, enhanced System-2 reasoning is integrated into System-1. We demonstrate how to instantiate it and how it helps to reason with visual inputs. Our method shows superiority in explainability, generalization, and efficiency in extensive experiments.

**Computation Cost.** Symbol predictions will increase the computational cost as a trade-off for explainability and generalization. It can be eased by discovering the hierarchical and reusable nature of the symbols, which is detailed in the supplementary.

**Broader Application.** The method could be more general and facilitate real-world applications. We choose activity understanding as a good and important initial test bed because it is a difficult task with complex visual patterns and a compositional nature. To verify the broader application, we provide some initial results on object recognition and VCR [36] tasks in the supplementary.

**Limitation.** Rules generated from language models mostly boost System-1, but sometimes bring language bias, *e.g.*, the person only decides to buy an orange despite apples in the background. Further, the approximation is mainly based on pre-trained LLMs and can be improved with more elaborate designs, *e.g.*, human-in-the-loop, customized LLMs with higher-quality knowledge. Besides, System-2 reasoning may be boosted if integrated into the System-1 training process. Also, instance-level activities and temporal encoding are beyond our scope since our main focus is on an overall framework to formulate and construct a novel symbolic system. We leave them to future work.

## Acknowledgments and Disclosure of Funding

Some of the drawing materials in Fig. 1,2 are designed by Layerace / Freepik.

This work is supported in part by the National Natural Science Foundation of China under Grants 62306175, National Key R&D Program of China (No.2021ZD0110704), Shanghai Municipal Science and Technology Major Project (2021SHZDZX0102), Shanghai Qi Zhi Institute, Shanghai Science and Technology Commission (21511101200).

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
