# Supplementary for Symbol-LLM: Leverage Language Models for Symbolic System in Visual Human Activity Reasoning

**Xiaoqian Wu**
Shanghai Jiao Tong University
enlighten@sjtu.edu.cn

**Yong-Lu Li**[*]
Shanghai Jiao Tong University
yonglu_li@sjtu.edu.cn

**Jianhua Sun**
Shanghai Jiao Tong University
gothic@sjtu.edu.cn

**Cewu Lu**[*]
Shanghai Jiao Tong University
lucewu@sjtu.edu.cn

## Contents

---

[*]Corresponding authors.

37th Conference on Neural Information Processing Systems (NeurIPS 2023).

**5   Boarder Application**           **12**

## 1   Notations

In Tab. 1, we conclude the notations in this work for clarity.

| Notation | Definition |
|---|---|
| $r$ | A rule. It takes the form $r : m_1 \wedge m_2 \wedge \ldots \wedge m_{n_r} \models c$. |
| $M = \{m_i\}_{i=1}^{n_r}$ | The premise symbols set of the rule $r$. |
| $c$ | The conclusion symbol of the rule $r$. |
| $n_r$ | The size of the premise symbols set $M$. |
| $\wedge$ | Logic AND. |
| $\vee$ | Logic OR. |
| $\models$ | Entailment. |
| $(\mathcal{S}, \mathcal{R})$ | The activity symbolic system. $\mathcal{S}$ is the symbol set, and $\mathcal{R}$ is the rule set. |
| $(\mathcal{S}_c, \mathcal{R}_c)$ | The activity symbolic sub-system given a conclusion $c$. |
| $A \backslash B$ | The set difference of A and B. |
| $\mathcal{D}$ | A very large-scale activity images database. $\mathcal{D} = \{(I, A, \mathcal{S}_I)\}$. |
| $I$ | An image. |
| $A_I$ | The activities happening in an image $I$. |
| $\mathcal{S}_I$ | Ongoing symbols in an image $I$. |
| $\mathcal{L}$ | An LLM. |
| $\mathcal{A}$ | The activity set contains multiple activity classes. $\mathcal{A} = \{A\}$ |
| $\mathcal{C}$ | The conclusion set contains multiple conclusions. $\mathcal{A}$ and $\mathcal{C}$ is equivalent. |
| $\mathcal{S}_A$ | The premise symbols set for activity $A$. It is implied from an LLM. |
| $e$ | The entailment score of a rule. |
| $e_h$ | The entailment score threshold to accept/reject a rule. We set $e_h = 0.9$. |
| $\tau(\cdot)$ | A function to measure the entailment score $e = \tau(r)$ of a rule $r$. |
| $r^{(j)}$ | The $j$-th rule in $\mathcal{R}_c$. It takes the form $r^{(j)} : m_1^{(j)} \wedge m_2^{(j)} \wedge \ldots m_{n_{r^{(j)}}}^{(j)} \models c$. |
| $m_i^{(j)}$ | The $i$-th premise symbol in the $j$-th rule of $\mathcal{R}_c$. |
| $p_{y,i}^{(j)}, p_{n,i}^{(j)}$ | The output probability of answering "Yes""No" for the symbol $m_i^{(j)}$ from a VLM. |
| $p_i^{(j)}$ | The probability of the symbol $m_i^{(j)}$. |
| $N_{query,c}$ | The number of queries to generate the symbolic sub-system given the conclusion $c$. |
| $N_{ent}$ | The number of sampling to calculate the entailment score. $N_{ent} = 5$. |
| $|M^{(j)}|$ | The size of the premise symbol set for a rule $r^{(j)}$. |
| $\{m_{i,k}^{(j)}\}_{k=1}^5$ | The paraphrased variants for the symbol $m_i^{(j)}$. |
| $\{p_{i,k}^{(j)}\}_{k=1}^5$ | The probability of the symbols $\{m_{i,k}^{(j)}\}_{k=1}^5$. |
| $std_i^{(j)}$ | The standard deviation of $\{p_{i,k}^{(j)}\}_{k=1}^5$. |
| $\mathcal{S}_{sys1}$ | The prediction of a System-1-like method. |
| $\mathcal{S}_{sys2}$ | The prediction of a System-2-like method. |
| $\mathcal{S}_{int}$ | The final integrated prediction. |
| $\alpha_1, \alpha_2$ | The re-scaling factors to normalize $\mathcal{S}_{sys1}, \mathcal{S}_{sys2}$. |

Table 1: Notations and their definition of this work.

**About the entailment notation.** We replace the notation "$\rightarrow$" in HAKE with "$\models$" for mathematical rigor. The notations "$\models$" (semantic entailment) and "$\rightarrow$" (logical/syntactic entailment) both describe the concept of one statement leading to another, but their emphasis is different. Semantic entailment focuses on the reasoning relationship based on truth values or interpretations. When all interpretations that make all formulas in $A$ true also make $B$ true, we say that $A$ semantically entails $B$. Syntactic entailment focuses on the ability to derive a conclusion from a set of premises based on formal reasoning rules. When $Q$ can be derived from the formulas in $P$ solely through reasoning rules, we say that $P$ syntactically entails $P$. Semantic entailment focuses on truth tables or interpretations (*i.e.*, under what circumstances or "worlds" a proposition is true), while syntactic entailment focuses on

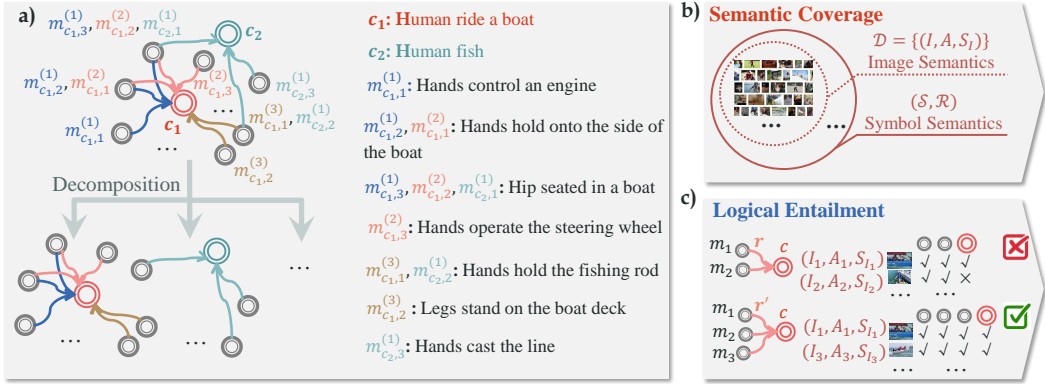

Figure 1: Our activity symbolic system. a) The structure and decomposition. Its two important properties: b) semantic coverage and c) logical entailment. It details part of Fig. 2 in the main text.

formal reasoning rules and proof processes. Visual reasoning focuses on interpretations instead of syntactic proof. Thus we replace the notation "→" in HAKE with "⊨".

## 2 More Details of Symbolic System Formulation

### 2.1 Explanation of the Illustration

In the main text, Fig. 2 illustrates the overview of the proposed activity symbolic system. We give a more detailed explanation here. This section focuses on the part of Fig. 2 related to the symbolic system formulation. The other parts are left to Sec. 3.1.

Fig. 1a explains the structure and decomposition of our activity symbolic system. The premise symbols (gray circles), the conclusion symbols (red and green circles), and the rules (blue, red, brown, and green edges) are shown. Example meanings of symbols are also listed. When decomposing the activity symbolic system into sub-systems, unrelated symbols/rules are removed within each sub-system. Fig. 1b explains the semantic coverage property, *i.e.*, comparing the image semantics in the database $\mathcal{D}$ and symbol semantics in the symbolic system $(\mathcal{S}, \mathcal{R})$. Fig. 1c explains the logical entailment property. The rule $r : m_1 \wedge m_2 \wedge m_3 \models c$ satisfies logical entailment, while $r' : m_1 \wedge m_2 \models c$ fails because in $(I_2, A_2, S_{I_2})$, $\{m_1, m_2\} \subset S_{I_2}$ but $c \notin A_2$.

### 2.2 Asymmetry of Rule

In Sec. 3.1.1 in the main text, we analyze the structure of the symbolic system as a type of directed hyper-graph instead of the indirect hyper-graph. This is because of the asymmetry of the rule: within one rule, the premises symbol and the conclusion symbol cannot swap positions. That is, for a rule $r : m_1 \wedge m_2 \wedge \ldots \wedge m_{n_r} \models c,$, another rule $r' : m_1 \wedge \ldots \wedge m_{i-1} \wedge c \wedge m_{i+1} \ldots \wedge m_{n_r} \models m_i, (1 \leq i \leq n_r)$ is not necessarily satisfied. For example,

$r$:  Hip seated in a boat $\wedge$ Hand hold onto the side of the boat $\wedge$ Hand operate the steering wheel $\models$ Human ride a boat

$r'$: Hip seated in a boat $\wedge$ Hand hold onto the side of the boat $\wedge$ Human ride a boat $\models$ Hand operate the steering wheel

The rule $r'$ is not satisfied since there are other possible conclusions, *e.g.*, "Hand controls the engine". Based on the asymmetry property of the rule, the activity symbolic system is defined based on a directed hyper-graph, with the edges pointing from the premise symbols to the conclusion symbol.

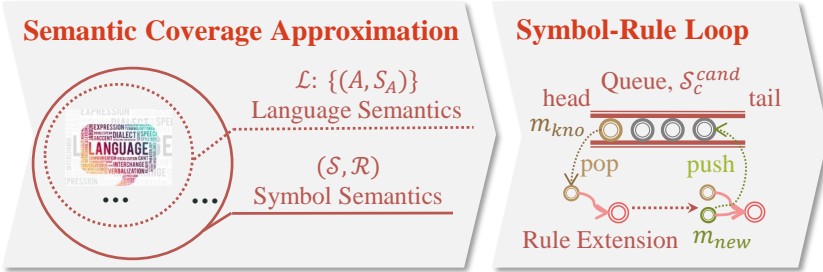

Figure 2: Semantic coverage can be approximated based on LLMs' knowledge and achieved via the symbol-rule loop. It details part of Fig. 2 in the main text.

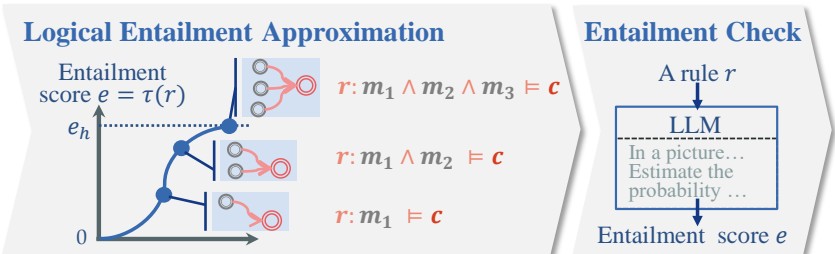

Figure 3: Logical entailment can be approximated based on an entailment scoring function and achieved by entailment check. It details part of Fig. 2 in the main text.

## 2.3 Structure of the Previous Symbolic System

Existing methods [12, 10] can also be formulated under the structure in Def. 3 in the main text. Their symbolic system is a specific example of Def. 3, where the premise set $M$ for each rule $r$ is the subset of the hand-crafted activity primitives, *i.e.*, $M \subset M^*$, $|M^*| = 76$ and $M^*$ is a fixed set.

# 3 More Details of Symbolic System Instantiation

## 3.1 Explanation of the Illustration

We further explain the other part of Fig. 2 in the main text. As is shown in Fig. 2, semantic coverage can be approximated by comparing the symbol semantics in the symbolic system $(\mathcal{S}, \mathcal{R})$ with the LLM semantics implying the premise symbols set $S_A$ for an activity $A$. In each symbol-rule loop, a known symbol $m_{kno}$ is taken from the queue $\mathcal{S}_c^{cand}$, and the rule extension is applied with a new symbol $m_{new}$ added. As is shown in Fig. 3, the logical entailment is measured via an entailment scoring function $\tau(\cdot)$. For the rule $r : m_1 \wedge m_2 \wedge m_3 \models c$, the entailment score for $m_1 \models c, m_1 \wedge m_2 \models c, m_1 \wedge m_2 \wedge m_3 \models c$ climbs up because more premise symbols are added. Once the entailment score $e$ surpasses the threshold $e_h$, it is regarded as a rule equipped with logical entailment and updated into the symbolic system. To implement the scoring function based on an LLM, we rewrite a rule $r$ as a sentence and design prompts.

## 3.2 Summarized Pipeline

Alg. 1 in the main text shows how to get the sub-system $(\mathcal{S}_c, \mathcal{R}_c)$ for a given conclusion $c$. We again show it in this supplementary and explain it for clarity. The target is to determine the rule set $\mathcal{R}_c = \{r^{(j)}\}_{j=1}^{N_c}$, where a rule takes the form $r^{(j)} : m_1^{(j)} \wedge m_2^{(j)} \wedge \ldots m_{n_{r^{(j)}}}^{(j)} \models c$, or equivalently, $r^{(j)} : \bigwedge_{m \in M^{(j)}} m \models c$. Correspondingly, the symbol set is $\mathcal{S}_c = (\bigcup_{j=1}^{N_c} M^{(j)}) \cup \{c\}$.

First, the initial symbols $\mathcal{S}_c^0$ are generated (L1 in Alg. 1). For the symbol-rule loop (L4-14), a queue $\mathcal{S}_c^{cand}$ stores candidate symbols, and each loop processes one symbol $m_{kno}$ in $\mathcal{S}_c^{cand}$. In each loop, a known symbol $m_{kno}$ is taken as an element in the premise symbol set $M$, and the rule extension

---
**Algorithm 1** Instantiating the Symbolic System
---
**Input:** conclusion $c$, entailment threshold $e_h$
**Output:** Symbolic Sub-System $(\mathcal{S}_c, \mathcal{R}_c)$
 1: $\mathcal{S}_c^0 \leftarrow$ Symbol_Initialization $(c)$
 2: $\mathcal{S}_c^{cand}.push(\mathcal{S}_c^0)$          $\triangleright$ A queue $\mathcal{S}_c^{cand}$ stores symbols
 3: $\mathcal{S}_c \leftarrow \{c\}, \mathcal{R}_c \leftarrow \{\}$
 4: **while** not $\mathcal{S}_c^{cand}.is\_empty()$ **do**
 5:    $m_{kno} \leftarrow \mathcal{S}_c^{cand}.pop()$     $\triangleright$ A known symbol $m_{kno}$ is taken for rule extension
 6:    $M = \{m_{kno}\}$        $\triangleright$ The premises set $M$ is set for the current rule
 7:    **while** Entailment_Check$(M) < e_h$ **do**
 8:     $m_{new} \leftarrow$ Rule_Extension$(M, c)$
 9:     $M = M \cup \{m_{new}\}$        $\triangleright$ A new symbol $m_{new}$ is added
10:    **end while**
11:    $\mathcal{S}_c \leftarrow \mathcal{S}_c \cup M$        $\triangleright$ The symbol set $\mathcal{S}_c$ is updated
12:    $\mathcal{R}_c \leftarrow \mathcal{R}_c \cup \{r : \bigwedge_{m \in M} m \models c\}$    $\triangleright$ The rule set $\mathcal{R}_c$ is updated
13:    $\mathcal{S}_c^{cand}.push(M \backslash \mathcal{S}_c)$    $\triangleright$ $M$ is added to $\mathcal{S}_c^{cand}$ with redundant symbols removed
14: **end while**
15: **return** $(\mathcal{S}_c, \mathcal{R}_c)$
---

is repeatedly applied with the new symbol $m_{new}$ added to $M$ until the entailment check is passed (L7-10). Thus, a rule $r : \bigwedge_{m \in M} m \models c$ which satisfies logical entailment is obtained and added to the rule set $\mathcal{R}_c$ (L12). Meanwhile, the new symbols are added to the symbol set $\mathcal{S}_c$ and the candidate symbol set $\mathcal{S}_c^{cand}$, and the repeated symbols are removed. The symbol-rule loop continues until $\mathcal{S}_c^{cand}$ is emptied. Finally, we get the symbol set $\mathcal{S}_c$ and rule set $\mathcal{R}_c$ for the conclusion $c$.

### 3.3 Determining Prompts

The prompts are determined via a trial-and-error process by human experts on randomly sampled 10 activities. Human experts have read the image samples of these activities, and thus have expert knowledge about what symbols/rules are more suitable for visual reasoning. Each round in the trial-and-error process is as follows:

- one expert writes a prompt
- an LLM generates answers by the prompt
- all experts judge whether the answer is suitable
- all experts modify the prompt based on the discovered problems

An example of the trial-and-error process of the prompts for symbol initialization is shown below.

1. Round1:
   - Prompt: In a picture, a person is boarding an airplane. What are useful visual features to distinguish his activity?
   - Answer: Useful visual features...could include: 1. Luggage: Look for the presence of suitcases, backpacks, or other types of bags that suggest the person is preparing to travel...
   - Comment: Object-centric, difficult to extract key information about human activities.

2. Round2:
   - Prompt: In a picture, a person is boarding an airplane. What is he doing?
   - Answer: The person in the picture is boarding an airplane. Boarding refers to the process of entering the aircraft before the flight begins. It typically involves presenting a valid boarding pass, showing identification, and walking through the designated gate or bridge to reach the plane.
   - Comment: Verbose.

3. Round3:

- Prompt: In a picture, a person is boarding an airplane. What is he doing? Answer with 5 concise, highly related phrases. The format is "\<human\> \<verb\> \<object\>".
- Answer: Person boarding airplane.
- Comment: The answer repeats the question.

4. Round4:
   - Prompt: In a picture, a person is boarding an airplane. What are his hands doing? Answer with 5 concise, highly related phrases. The format is "\<hands\> \<verb\> \<object\>".
   - Answer: 1. Hands holding a boarding pass. 2. Hands placing luggage in overhead compartment. 3. Hands adjusting seatbelt. 4. Hands waving goodbye to loved ones. 5. Hands gripping a luggage handle.
   - Comment: Satisfactory.

## 3.4 An Example of Instantiation

We show the detailed process of generating the symbolic sub-system for the activity "`human board an airplane`". It is an extension of the results in Fig. 4, Tab. 1 in the main text, and the method described in Sec. 3.2 in the main text. The LLM used here is OpenAI GPT3.5 via API, where the role is set as "You are helping me understand human activities in a picture.".

To generate initial premise symbols (L1 in Alg. 1), we prompt as:

```
(Symbol Initialization)
In a picture, a person is boarding an airplane.  What are his hands
doing?  Answer with 5 concise, highly related phrases.  The format is
"<hands> <verb> <object>".  Output Format:  1.  xxx 2.  xxx 3.  xxx 4.
xxx 5.  xxx
```

Then 5 initial symbols are generated from the answer, *e.g.*, "`hold a boarding pass`", "`place luggage in overhead compartment`".

Then, we extend the symbol "`hold a boarding pass`" into a rule. To add a new premise symbol (L8 in Alg. 1), we prompt as:

```
(Rule Extension)
In a picture, there is an airplane.  IF [The person is holding a
boarding pass.]  AND [condition] THEN [The person is boarding the
airplane.].  [condition] is one concise phrase.  The format is
"<The person's hands/arms/hip/legs/feet> <verb> <object>".  What is
[condition]?  Output Format:  [condition] is:  [xxx].
```

With an answer "[condition] is: The person is walking towards the boarding gate", we get the new symbol "`walk towards the boarding gate`" and a candidate rule "`hold a boarding pass` $\wedge$ `walk towards the boarding gate` $\models$ `Human board an airplane`".

Then, an entailment check is applied for the candidate rule (L7 in Alg. 1). The prompt is:

```
(Entailment Check)
In a picture, there is an airplane.  The person is holding a boarding
pass.  The person is walking towards the boarding gate.  Estimate
the probability that he is boarding the airplane at the same time.
Choose from:  (a) 0.1, (b) 0.5, (c) 0.7, (d) 0.9, (e) 0.95, (f) unknown.
Output Format:  a/b/c/d/e/f.
```

We sample the answer 5 times and take the average entailment score. If the score $e < e_h$, we repeat the rule extension and entailment check.

```
(Rule Extension)
In a picture, there is an airplane.  IF [The person is holding
a boarding pass.  The person is walking towards the boarding
gate.]  AND [condition] THEN [The person is boarding the airplane.]
[condition] is one concise phrase.  The format is "<The person's
hands/arms/hip/legs/feet> <verb> <object>".  What is [condition]?
Output Format:  [condition] is:  [xxx].
```

With a new symbol "`luggage visible beside him`", the entailment score of the new rule is measured via prompting:

(Entailment Check)
```
In a picture, there is an airplane.  The person is holding a boarding
pass.  The person is walking towards the boarding gate.  The luggage is
visible beside him.  Estimate the probability that he is boarding the
airplane at the same time.  Choose from:  (a) 0.1, (b) 0.5, (c) 0.7, (d)
0.9, (e) 0.95, (f) unknown.  Output Format:  a/b/c/d/e/f.
```

The rule extension and entailment check are repeated until the entailment score threshold is reached. Finally, we get a rule

```
hold a boarding pass ∧ walk towards the boarding gate ∧ luggage visible
beside him ∧ boarding pass is scanned by airport staff ⊨ Human board an
airplane
```

The prompt differences are small modifications necessary to adapt different dataset/task settings. When an interacted object is known, it is listed in the rule extension prompt for emphasis (*e.g.*, "there is an airplane"). For HAKE-PaSta, HOIs are known as conditions, and the symbol initialization is omitted because the known HOIs are initial premise symbols for rule extension.

## 3.5   Computation Cost

Given a conclusion $c$, the number of query $N_{query}$ with a LLM is:

$$N_{query,c} = 1 + N_{ent} * \sum_{j=1}^{N_c} |M^{(j)}|, \tag{1}$$

where 1 refers to the first query to generate initial symbols, $N_{ent} = 5$ is the number of sampling to calculate the entailment score, and $|M^{(j)}|$ is the size of the premise symbol set for a rule $r^{(j)}$.

For the example shown in Fig. 4 and Tab. 1 in the main text, we have $N_{query} = 1 + 5 * (4 + 2 + 3 + 3 + 2 + 4 + 3 + 2 + 2 + 3 + 3 + 3 + 3 + 3 + 3) = 216$, where the early stop symbols "10""14" also need several queries to determine early stop. The exploited knowledge is reusable to facilitate future study.

## 3.6   Comparing with HAKE

To compare the difference of the symbolic system between our method and HAKE [10], we list some of the rules for the activity "`human buy an orange`" in Tab. 2. We can see that our symbolic system is superior in semantic coverage and rule rationality.

## 3.7   Performance Evaluation

To evaluate the symbolic system, we construct a SymAct (Symbol Activity) test set. It is a small subset of the HICO test set (120 images for 10 activity classes).  The 10 activity classes are: "human board/wash an airplane", "human park/repair a bicycle", "human feed/race a horse", "human break/sign a baseball bat", and "human buy/wash an orange". These activities have relatively complex visual patterns and cover different interacted objects. For each image and each of its ground-truth activities, we list all of the symbols related to the activity in the symbolic system and annotate whether a symbol happens (0/1) in the image via human judgment.

We find the proposed symbolic system has: **1) broader semantic coverage** than HAKE. For quantitative measurement, we count the happening symbols for each image-activity pair under different symbolic systems.  The average number is 10.8 for ours and 1.8 for HAKE. Though a rough estimation, the gap in symbol counts indicates that the symbol semantic of HAKE is limited. Furthermore, in our symbolic system, different activities have different symbols, further boosting semantic coverage. **2) more rational rules** than HAKE. We evaluate confusion by counting different image-activity pairs which share the same symbols. For the HAKE symbolic system, the confusion problem is severe, with 261 confusion pairs over $C_{120}^2 = 7140$ pairs (accounting for 3.7%). For our

symbolic system, there are no confusion pairs on the test set because of the presentation ability of symbols and entailment check.

| talk with seller | ∧ | reach for an orange | - | - | | | ⊨ buy an orange |
|---|---|---|---|---|---|---|---|
| talk with seller | ∧ | seller hand over orange | - | - | | | ⊨ buy an orange |
| stand in front of fruit stand | ∧ | place orange in a bag | ∧ | pick orange from a basket | | | ⊨ buy an orange |
| stand in front of fruit stand | ∧ | hold a bag of oranges | ∧ | reach for a wallet | | | ⊨ buy an orange |
| stand in front of fruit stand | ∧ | seller put the orange in bag | ∧ | give money to seller | | | ⊨ buy an orange |

(a) Rules from our proposed symbolic system.

| head: talk to | - | - | - | - | - | - | ⊨ buy an orange |
|---|---|---|---|---|---|---|---|
| hand: reach for | ∧ | head: inspect | - | - | - | - | ⊨ buy an orange |
| hand: hold | ∧ | head: smell | - | - | - | - | ⊨ buy an orange |
| talk with seller | ∧ | reach for an orange | - | - | - | - | ⊨ buy an orange |
| hand: hold | ∧ | hand: reach for | ∧ | hand: squeeze | ∧ | head: drink with | ⊨ buy an orange |

(b) Rules from the open-source code of HAKE [10].

Table 2: Rules for the activity "`human board an airplane`" from the proposed symbolic system and HAKE [10].

## 3.8 Robustness

We analyze the robustness of the generated symbolic system as follows.

**Convergence.** The symbol-rule loop should find a symbolic sub-system that "converges". In other words, after a certain stopping criterion, a new round will bring few or even no new symbols to the sub-system. In practice, the stopping criterion is: At most 15 symbols are used for rule extension to avoid semantic redundancy. After that, new symbols are not used for rule extension, leading to a stop of the loop. With the stopping criterion, the symbol-rule loop satisfies convergence. Take Tab.1 as an example: in rounds 1-5, the newly added symbols count is 5, 5, 3, 3, 1, which shows a convergence trend. Fig.5(a) in the main text shows a similar convergence trend. More generally, for the randomly sampled 50 activities, a new round brings an average of 1.9 symbols to the sub-system, verifying the convergence.

**Sensitivity to initial conditions.** The symbol-rule loop should provide a reliable and stable symbolic sub-system over a range of initial conditions. The sensitivity to initial conditions mainly comes from the asymmetry of rule generation: with 4 as an initial symbol, a rule connecting 4&10&6 is generated, while the rule sometimes cannot be generated with 10 as an initial symbol. However, we find such sensitivity an infrequent case with multiple rules sampling. To verify this, we randomly sample 15 different initial symbol sets to generate the sub-system and then measure the average pairwise graph similarity. The average similarity on 10 randomly sampled activities is 0.91, verifying the low sensitivity to initial conditions.

**Sensitivity to different prompts.** Experiment results are sensitive to prompts with major differences. Using the older version prompt during the trial-and-error process brings difficulty to correct and automated symbol generation. Experiment results are less sensitive to prompts with minor differences, *i.e.*, slightly paraphrasing the prompts. We conducted a brief experiment: We get another 2 paraphrased symbol init prompts and another 2 paraphrased rule extension prompts. We use these $3 * 3 = 9$ prompts to generate 9 sub-systems and measure the average pairwise graph similarity. The average similarity (range: [-1,1]) on 10 randomly sampled activities is 0.83, verifying the relatively low sensitivity to different prompts.

## 4 More Details of Visual Reasoning

### 4.1 Visual Symbol Extractor

The visual symbol extractor is based on a VLM BLIP2 [8] with its language model used as a scoring function [1]. A language model represents a distribution over potential completions $p(w_k|w_{<k})$, where $w_k$ is a word that appears at a $k$-th position in a text. While typical generation applications

sample from this distribution, we can also use the model to score a candidate completion selected from a set of options [1]. With a symbol converted into a sentence, *e.g.*, "The person's hip is seated in a boat. Yes/No?", the language model outputs the probabilities of constrained responses "Yes""No" as $p_{y,i}^{(j)}, p_{n,i}^{(j)}$. Then the probability $p_i^{(j)}$ of the symbol can be obtained via normalizing the Yes/No answer with softmax function as:

$$p_i^{(j)} = \frac{e^{p_{y,i}^{(j)}}}{e^{p_{y,i}^{(j)}} + e^{p_{n,i}^{(j)}}}. \tag{2}$$

### 4.2 Visual Symbol Checker

To check symbols, one symbol $m_i^{(j)}$ is paraphrased [3] into 5 variants $\{m_{i,k}^{(j)}\}_{k=1}^5$. The predictions of variants should be similar due to semantic consistency. Thus, the standard deviation $std_i^{(j)}$ of the predictions $\{p_{i,k}^{(j)}\}_{k=1}^5$ is calculated. In practice, the symbols $m_i^{(j)}$ with $std_i^{(j)} \geq 0.05$ are regarded as uncertain symbols and filtered out. The proportion of filtered symbols is around 5%.

### 4.3 Dataset and Metric

We conduct experiments on image-level activity understanding benchmarks with diverse tasks.

**HICO** [2] is a Human-Object Interaction (HOI) recognition benchmark, with 38,116 / 9,658 images in the train/test set. It defines 600 HOIs composed of 117 verbs and 80 COCO objects [13].

**Stanford40** [18] is an activity recognition dataset, with 9,532 images for 40 actions. Its train/test set contains 4,000/5,532 images. We omit its bounding box annotations and focus on the image-level recognition task.

**HAKE** [12, 11] provides 118K+ images, which include 285K human instances, 250K interacted objects, and 724K HOI pairs with human body part states (PaSta) [12, 14]. To utilize the abundant activity samples, we follow the setting in [9] to split HAKE [12], with 22,156 images in the test set. Differently, we design different tasks to facilitate activity reasoning.

**HAKE** [12, 11]**-Verb** is a verb recognition benchmark that defines 156 verbs related to human activity. To remove ambiguity, the verb prediction is conditioned upon ground-truth object labels, *i.e.*, excluding verbs contradictory to known objects.

**HAKE** [12, 11]**-PaSta** is a multi-choice QA benchmark to predict PaSta with the ongoing HOIs known as conditions, dealing with finer-grained human activity. To maintain the sample balance of PaSta, we select a subset of 1,438 images from the original one [9] and match a QA pair for each image.

For HICO [2] and HAKE [12, 11]-Verb, we use mAP for multi-label classification. For Stanford40 [18], we use mAP following the original setting. For HAKE [12]-PaSta, we use top-1 accuracy for multi-choice QA.

For HICO [2], HAKE [12]-Verb and Stanford40 [18], symbols and rules are generated in the summarized pipeline in Sec. 3.2.3 in the main text and Sec. 3.2 in supplementary.

### 4.4 Results and Analysis

We further analyze the visual reasoning results in Tab. 2 in the main text.

**More baselines.** We omit some HICO [2] baselines in the main text because of space limitations. We provide them here in Tab. 3.

**Integrating System-1/2-like methods.** As a plug-and-play, the reasoning is compatible with existing System-1-like methods. We follow the setting of HAKE [10] to integrate the System-2 reasoning result $\mathcal{S}_{sys2}$ with the prediction $\mathcal{S}_{sys1}$ of System-1-like methods. The final prediction is $\mathcal{S}_{int} = \alpha_1 * \mathcal{S}_{sys1} + \alpha_2 * \mathcal{S}_{sys2}$, where $\alpha_1, \alpha_2$ is a re-scaling factor to normalize $\mathcal{S}_{sys1}, \mathcal{S}_{sys2}$ and calculated from $\mathcal{S}_{sys1}, \mathcal{S}_{sys2}$.

**Performance gap between System-1-like methods.** We mainly choose 2 typical System-1-like methods: CLIP [17] and BLIP2 [8]. Comparing the two baselines, we find that with a frozen language

| Method | mAP | |
|---|---|---|
| | fine-tuned | zero-shot |
| R*CNN [6] | 28.50 | - |
| Girdhar *et al.* [5] | 34.60 | - |
| Mallya *et al.* [16] | 36.10 | - |
| Pairwise [4] | 39.90 | - |
| RelViT [15] | 40.10 | - |
| CLIP [17] | 67.12 | 37.08 |
| CLIP [17]+Reason | **69.73** | **43.21** |
| BLIP2 [8] | - | 50.61 |
| BLIP2 [8]+Reason | - | **53.15** |

Table 3: Visual reasoning results on HICO [2].

| Symbol probability | I | II |
|---|---|---|
| $m_1$: hold a boarding pass | 0.31 | 0.09 |
| $m_2$: place luggage in overhead compartment | 0.86 | 0.76 |
| $m_3$: adjust seatbelt | 0.47 | 0.97 |
| $m_4$: wave goodbye to loved ones | 0.22 | 0.09 |
| $m_5$: grip a luggage handle | 0.78 | 0.52 |
| $m_6$: walk towards the boarding gate | 0.21 | 0.05 |
| $m_7$: luggage visible beside him | 0.89 | 0.34 |
| $m_8$: boarding pass is scanned by airport staff | 0.86 | 0.07 |
| $m_9$: stand on the jet bridge | 0.54 | 0.44 |

| Symbol probability | I | II |
|---|---|---|
| $m_{10}$: luggage is loaded onto the plane | 0.73 | 0.77 |
| $m_{11}$: reach for the airplane door handle | 0.22 | 0.91 |
| $m_{12}$: stand in line with carry-on luggage | 0.11 | 0.04 |
| $m_{13}$: hold the carry-on luggage | 0.30 | 0.64 |
| $m_{14}$: open the airplane door | 0.34 | 0.91 |
| $m_{15}$: move forward in the line | 0.40 | 0.28 |
| $m_{16}$: move towards the airplane door | 0.10 | 0.72 |
| $m_{17}$: airline staff checking the boarding pass | 0.78 | 0.13 |

Figure 5: Two failure cases for the activity "human board an airplane". The rules are the same as those in the main paper Tab.1.

model, BLIP2 [8] is more capable of understanding activity semantics and outperforms CLIP [17] in various zero-shot benchmarks. Furthermore, the performance boost from reasoning is more evident on CLIP [17] than on BLIP2 [8]. For example, on HICO [2], visual reasoning improves 6.13 mAP for CLIP [17] while 2.54 mAP for BLIP2 [8]. The performance gap between System-1-like methods can be partly explained by the model ensemble: the visual symbol extractor in reasoning is based on BLIP2 [8], thus applying reasoning to CLIP [17] model brings in more bonus. In ablation studies, we analyze the effects of the model ensemble (Sec. 4.4, the Reasoning section in the main text). We find that trivially combining with baseline results (CLIP [17]+BLIP2 [8]) (44.76 mAP) outperform the baseline CLIP [17] (43.92 mAP), but integrating reasoning is superior (48.95 mAP). It verifies the necessity of System-2 reasoning other than trivially combining predictions from two models.

**Performance gap between benchmarks.** The performance improvement on HAKE [10]-PaSta is relatively smaller compared with other benchmarks. One possible explanation is the conclusion in HAKE [10]-PaSta is PaSta, which is more with more ambiguity compared with the activity class, *e.g.*, "hold something" vs. "ride a boat". The ambiguity of multi-choice answers hinders reasoning, but it is caused by the inherent ambiguity of the PaSta definition instead of the proposed reasoning method. However, despite the ambiguity, HAKE [10]-PaSta is still valuable in verifying the effectiveness of reasoning for finer-grained activity understanding tasks.

**Failure Cases.** We provide two failure cases (false negative) for the activity "human board an airplane" in Fig. 5. In image I, the failure is caused by the prediction error of symbol probabilities. Some symbols are predicted as low probabilities by mistake, *e.g.*, "walk towards the boarding gate". Tiny human bounding boxes possibly cause the symbol prediction error. In image II, the failure emerges because the generated rules do not cover the situation in the image, *i.e.*, errors of the symbolic system. For example, the image contains the symbol "climb up the airplane edge" which is not in the symbolic system.

## 4.5 Computation Cost

Symbol predictions will increase the computational cost as a trade-off for explainability and generalization. In this paper, we mainly focus on the effectiveness and implementation of our insight and do not preferentially consider efficiency. It can be eased by discovering the hierarchical and reusable nature of the symbols. We show some initial exploration below.

**Hierarchical structure.** Not all symbols need to be predicted via pruning. Symbols can be organized as a tree with hierarchical prediction. For the Fig.6 example in the main text, $m_1...m_{10}$ can be organized as:

- $m_a$: Interact with a seller:
    - $m_1$:Talk with seller
    - $m_3$: Seller hand over orange
    - $m_{10}$: Give money to seller

- $m_b$: Interact with the oranges:
    - $m_2$: Reach for an orange

- $m_c$: Interact with a container:
    - $m_5$: Place orange in a bag
    - $m_6$: Pick orange from a basket
    - $m_7$: Hold a bag of oranges
    - $m_9$: Seller put the orange in bag

- $m_d$: Payment Process:
    - $m_4$: Stand in front of fruit stand
    - $m_8$: Reach for a wallet

The father symbols (e.g., $m_a$) can be summarized based on the son symbols (e.g., $m_1$) via simple semantic extraction (*e.g*., LLM prompt). On the symbol tree, the probability of a son symbol is no higher than its father symbol. When the probability of a father symbol is lower than a threshold (e.g., in image III, $m_a$ probability less than 0.1), its son symbols ($m_1$, $m_3$, $m_{10}$) will be assigned the threshold directly without extra computation. Thus, the computational cost is saved via the hierarchical structure of symbols. For example, image III only needs 5 instead of 10 calculations: Interact with a seller(x), Interact with the oranges, Reach for an orange, Interact with a container(x), and Payment Process(x). The computation save will be more evident with the number of symbols scales.

**Reusable symbols.** The total number of symbols to measure will increase with the number of potential activity classes, but not scale linearly. It is unavoidable to measure probabilities of more than one activity class for activity understanding tasks. This is because activity recognition is a multi-label classification rather than a multi-class classification: A person typically performs multi-action simultaneously, *e.g*., standing while eating. Thus, the increase in symbol measurement comes from the inherent characteristics of activity tasks. However, the symbols are reused for different activity classes without repetitive computation. For example, "buy orange" and "visit store" shares the symbols: ($m_a$) Interact with a seller, ($m_d$) Payment process, ($m_1$) Talk with the seller, ($m_{10}$) Give money to the seller, ($m_8$) Reach for a wallet. It is also illustrated in the main paper Fig.2(a): the red and green activities share some symbols (gray), *e.g*., second from right, third from left.

**Quantitative comparison.** We provide a detailed analysis of a small case. There are 38 images known to contain oranges, where verbs (buy/cut/eat/hold/inspect/peel/pick/squeeze/wash) are to be predicted. We report the number of operations average per image on it. Without reasoning, activity classes are predicted directly, and the number of operations varies from images due to different ground-truth object labels which verb prediction conditions upon. The average number of operations is 9. With reasoning, the average number of operations is 71. However, with reusable symbols, the number is reduced to 31, and adding a hierarchical structure further reduces it to 23.

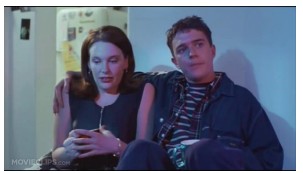

**What are person on the right doing?**
a)  person on the left is taking person on the right home.
**b) person on the right are on a first date.**
c) person on the left is conducting a job interview with
person on the right.
d) They are picking something up.

Figure 6: An example QA in VCR dataset.

# 5    Boarder Application

The method could be more general and facilitate real-world applications. We choose activity under-standing as a good and important initial test bed because it is a difficult task with complex visual patterns and a compositional nature. The proposed symbolic system can be generalized to various tasks (*e.g.*, classification, visual commonsense) with a similar formulation. Semantic coverage and logical entailment remain two important properties under a more general setting, and we can reason with visual inputs via a similar pipeline. We show two initial experiments below.

**Object classification.**  One example rule for an object (*e.g.*, crocodile) is:  long, slightly curved body $\wedge$ four short legs $\wedge$ ...  $\wedge$ a long, muscular tail $\rightarrow$ crocodile.  We conduct experiments on CIFAR-100 [7] test set with a zero-shot setting. The performance is 77.50% accuracy for CLIP and 78.31% accuracy for CLIP+Reasoning.

**Visual commonsense.** One example rule for the image and question in Fig. 6 is: person on the right shows affectionate gestures $\wedge$ person on the left engages in conversation $\wedge$ ... $\wedge$ The environment is a restaurant $\rightarrow$ b. We conduct experiments on VCR [19] val set. BLIP2 is adopted as a baseline as it is a VQA-style task. VCR val set has 26534 image-question pairs, where we select a subset of 557 pairs for the test. The selected pairs are highly related to human activities (instead of role, mental, *.etc*) and only involve *one* person's activity. The performance is 46.32% accuracy for BLIP2 and 47.08% accuracy for BLIP2+Reasoning.

We can see that the proposed pipeline can be applied to general visual tasks and facilitate reasoning. In this paper, we focus on method design instead of task generalization. We will analyze more general settings in future work.