# OpenReview forum: "Symbol-LLM: Leverage Language Models for Symbolic System in Visual Human Activity Reasoning"
_NeurIPS.cc/2023/Conference — NeurIPS 2023 poster_

### Official Review · Reviewer_yAvu · 2023-06-29

**Soundness:** 3 good
**Presentation:** 3 good
**Contribution:** 3 good
**Rating:** 6
**Confidence:** 2

**Summary:**

The paper addresses  activity reasoning, where the task is to detect human activities given visual inputs.
To achiever compositional generalization for activity understanding, the paper propose a new symbolic system that is capable of relational reasoning.
In the experiments, the proposed system improved naive CLIP models showing its strong reasoning capability.

**Strengths:**

Overall, the paper addresses an important problem for integrating symbolic reasoning with neural perception models.  The paper demonstrates an instantiation for such an integration by providing formal definitions.  The paper conveys its idea with a good presentation using clear visualizations. The obtained results are interesting.
These contributions are beneficial for the community to build agents that can perform complex reasoning and understand the world better beyond simple visual perception.

**Weaknesses:**

My major concerns regarding this paper are listed as follows.
- The notation and semantics of the proposed method are unclear.  The rule is denoted using the logical entailment $\models$. Is this different from implication $\rightarrow$, which is commonly used for logic?  In Tab. 2 in the appendix, $\rightarrow$ is used.  Since the definition of logical entailment is not provided in the paper, it is unclear how the rules can be understood as formulae. The implication and entailment articulate different concepts.
- Related to the first point, it is not clear how the proposed symbolic system is related to other symbolic reasoners, e.g. standard first-order logic? What are the pros and cons?
- The paper mainly addresses activity reasoning, but it is not well-described in the paper. Even though some visualizations are provided in the paper,  it is not easy to follow the arguments for readers who are not familiar with this topic. It would be beneficial to provide some simple examples throughout the paper.
- Related to the previous point, the empirical setting is not easy to follow. E.g. on line 232 , it says, "We follow the setting of HAKE [20] to integrate the reasoning ...", however, this explanation is not self-contained. The experimental settings should be articulated more clearer in the paper.

- Other minor issues:
	- Definition 5 is not clear. What are activity semantic and ongoing symbols?

**Questions:**

- The rule is denoted using the logical entailment $\models$. Is this different from implication $\rightarrow$, which is commonly used for logic?  (In Tab. 2 in the appendix, $\rightarrow$ is used.)
- How the proposed symbolic system is related to other symbolic reasoners, e.g. standard first-order logic? What are the pros and cons?

**Limitations:**

The authors listed some limitations in the paper.

---

> ### Author Rebuttal · Authors · 2023-08-09
>
> We thank the reviewer for the feedback and hope to address the questions and concerns below.
>
> ### Logic Notations
> The notations "$\models$" (semantic entailment) and "$\rightarrow$" (syntactic entailment) both describe the concept of one statement leading to another, but their emphasis is different.
>
> Semantic entailment focuses on truth tables or interpretations (i.e., under what circumstances a proposition is true), while syntactic entailment focuses on proof processes based on formal reasoning principles.
> Visual reasoning focuses on interpretations instead of syntactic proof. Thus we replace the notation "$\rightarrow$" in HAKE with "$\models$" for mathematical rigor. The "$\rightarrow$" in appendix Tab. 2 should be corrected as "$\models$".
> We will add explanations and correct notations in the final version.
>
>
> ### Literature Review of Symbolic Reasoners
>
> Compared with standard first-order logic:
>
> - We follow the principles of the standard first-order logic (FoL), e.g., logical connectives $\wedge, \vee$. The standard FoL is typically built upon a simplified world model, where symbols are definitely True/False. However, when adapting FoL in vision reasoning tasks, the symbols (e.g., $m_1,...m_{10}$ in Fig. 6) are not known to be True/False. Instead, the models are required to predict the symbol probabilities (usually via a neural network). Thus, a hybrid "neuro-symbolic" method should be adopted.
>
> Compared with other neuro-symbolic reasoners:
>
> - Pros: Compared with reasoners for visual question answering on simplified synthetic datasets [1], our method solves real-world visual activity understanding tasks that involve complex patterns (rules). Compared with other activity reasoning methods [2], we propose a novel symbolic system with broad semantics and rational rules to improve compositional generalization. Also, more explainable and lightweight calculations based on fuzzy-logic is provided.
> - Cons: We mainly focus on the design of a symbolic system, while symbol recognition is another important issue left for more exploration. Please refer to Reviewer zui3 "System-1/2 Ablation" and "Symbol Recognition Subtleties".
>
> We will add explanations in the "Related Work" section.
>
> [1] Yi, Kexin, et al. "Neural-symbolic vqa: Disentangling reasoning from vision and language understanding." Advances in neural information processing systems 31 (2018).
>
> [2] Li, Yong-Lu, et al. "Hake: a knowledge engine foundation for human activity understanding." IEEE Transactions on Pattern Analysis and Machine Intelligence (2022).
>
> ### Writing
>
> Thanks for your advice. We will provide more examples and clarify the empirical settings and definitions.
>
> The empirical setting (main paper L232) is detailed in Appendix L155-159. The final prediction is the weighted sum of two predictions with a calculated re-scaling factor.
>
> In Def-5, "activity semantics" are ground-truth activities happening in an image $I$, and "ongoing symbols" are ground-truth symbols happening in the image, which can be used as premises to reason out the activities. For example, in Fig.6 image-I, "activity semantics" include "human buy orange", and "ongoing symbols" include "talk with seller", "seller hand over orange", etc.

---

> > ### Comment · Reviewer_yAvu · 2023-08-14
> > **Reply to the rebuttal**
> >
> > I thank the authors for addressing the concerns adequately.
> >
> > Regarding the motivation of probabilistic reasoning, it would be beneficial to add the literature on probabilistic (logic) programming [1,2] since they essentially have the same motivation as explained in the rebuttal, i.e. dealing with uncertainties on symbolic reasoning. Describing the difference/novelty compared to them would clarify the contribution of the paper more.
> >
> >
> > [1] Luc De Raedt, Kristian Kersting, Sriraam Natarajan, David Poole: Statistical Relational Artificial Intelligence: Logic, Probability, and Computation. Synthesis Lectures on Artificial Intelligence and Machine Learning, Morgan & Claypool Publishers 2016
> >
> > [2] Daan Fierens, Guy Van den Broeck, Joris Renkens, Dimitar Sht. Shterionov, Bernd Gutmann, Ingo Thon, Gerda Janssens, Luc De Raedt: Inference and learning in probabilistic logic programs using weighted Boolean formulas. Theory Pract. Log. Program. 15(3): 358-401 (2015)
> >
> > As the concerns are addressed, I raise the score to 6 (from 5).

---

> > > ### Author Response · Authors · 2023-08-14
> > > **Thank you for your feedback!**
> > >
> > > We thank the reviewer for the feedback!
> > > We will add the literature on probabilistic (logic) programming [1,2] and describe the difference/novelty compared to them:
> > >
> > > The standard probabilistic (logic) programming typically solves simplified relational tasks, e.g., the WebKB dataset, and the BoxWorld problem. In comparison, we focus on a real-world vision task that involves complex patterns (rules). Thus, a novel symbolic system has been proposed.

---

### Official Review · Reviewer_sPA9 · 2023-07-09

**Soundness:** 2 fair
**Presentation:** 3 good
**Contribution:** 3 good
**Rating:** 5
**Confidence:** 3

**Summary:**

This work is in the area of human visual activity reasoning and tries to improve System-2 (logic, deliberative reasoning) integrated with System-1 (that deals with intuitivity, associativity, and correlation) to improve explainability, generalization and efficiency which is demonstrated through empirical evaluation. This work attempts to improve over existing methods which use limited symbols, and are therefore is insufficient to cover complex patterns of activities for visual activity understanding. They do this by proposing a new system with broad-coverage symbols and rational rules.

**Strengths:**

-This work attempts to solve the lack of compositional generalization (generalized understanding of visual concepts, their relationships, and novel combinations) by creating a system of symbols with broad semantic coverage and rules that are rational and unambiguous. E.g., an existing work contains the following rule:
Hand hold ^ Feet close with |= human ride a boat.
This rule does not cover the complex patterns of human activities.

- This work improves the existing methods by using an inexpensive annotation method by utilizing LLMs (OpenAI GPT3.5 API). This method is able to generate diverse symbols and rules quickly as shown in their symbolic system experiment. For the semantic coverage of the huge image dataset (the target domain), they replace it (the target domain) with the semantic coverage of language models.

**Weaknesses:**

The core message of the paper appears to be that more symbols with greater coverage leads to improved performance.  Figure 4 provides a high-level view of how these symbols are generated from LLM output, and Figure 6 notionally shows how they integrate with visual processing, but technical details are missing or unclear. For instance, how are initial symbols "extended into rules satisfying entailment"?  In what format are symbols and rules stored?  What is the method for integrating the symbolic reasoning system into the overall visual processing pipeline?

Their method uses symbol-rule loop to generate new rules whereby they use a known symbol to generate a new symbol and connect it with the rules. The author(s) do not present examples to test the robustness of this method.

**Questions:**

Most questions given above in Weaknesses.

From Fig 6, it is not clear if the method will generate the correct set of rules if the images also contained say apples in various circumstances in the background, but the person only decides to buy orange.  Can you please clarify this?

**Limitations:**

One limitation regarding symbolic representations of continuous activities is mentioned.  The authors have not addressed negative societal biases, and I can think of cases where some societal biases might arise.  Activity-based reasoning is at its core a kind of "stereotype"-based reasoning: "Balls are round.  Balls roll.  Apples are usually round.  Apples probably roll." In the domain of common objects, the risks of such stereotype-based reasoning are probably minimal, although if other objects with potentially harmful affordances (e.g., firearms or other weapons) are used without noting that those behaviors are harmful and should be avoided, this presents potential for abuse.  Can things like race, gender, or social class be represented symbolically? Should they?  Would symbolic representation of these bias system outputs in the presence of "coded" objects like firearms?  Beyond universal basic behaviors, who decides what the symbolic representations of them are? Without attending to these questions and more, there is a risk of attributing "typical" or "stereotypical" behaviors to certain individuals or groups of people, and risk generalizing them to other individuals, and hence reproducing patterns of bias and discrimination.

---

> ### Author Rebuttal · Authors · 2023-08-09
>
> We thank the reviewer for the feedback and hope to address the questions and concerns below.
> ### Method Details Explanation
>
> Thanks for your advice. We will add more explanations in the final version.
>
> 1. Rule Extension
>     - It is explained in the main paper Sec. 3.2.3 and Alg. 1 (L6-10). The initial symbols are extended into rules by adding one new symbol and checking entailment in an interleaved way.
>
> 2. Format of Symbols\&Rules
>     - Symbols are stored as natural language descriptions, and rules are stored as a premise symbol set and a conclusion symbol.
> It is explained in the rule examples (L45) and the symbolic system definition (Def.3).
>
> 3. Visual Reasoning
>     - It is explained in Sec. 3.3 and Fig. 3, where the activity prediction is calculated based on the rules of the symbolic system and the probabilities of the symbols.
>
> 4. Rule in Fig. 6
>     - We analyze whether the method will make errors when there are contexts (apple) disturbing the activity prediction (human buy orange) in the image.
>     - The unrelated contexts are less likely to hinder the accuracy of the generated rules because the entailment check guarantees the rationality of the rule. For the activity "human buy orange", there are symbols explicitly pointing out the existence of "oranges" and the interaction with "oranges" instead of "apples" (e.g., $m_7$: hold a bag of oranges).
> When there are apples in the background, the reasoning will focus on whether the person is really "holding" a bag of oranges.
> Therefore, the correct set of rules is generated.
>
>
> ### Symbol-Rule Loop Robustness
>
> Fig. 4 and Tab. 1 present an example of rule generation loops and reveal some phenomena, e.g., repeated rules, and early stop of rule extension.
> We further analyze robustness as follows. Additional analysis and results will be updated.
>
> 1. Convergence. The symbol-rule loop should find a symbolic sub-system that "converges". In other words, after a certain stopping criterion, a new round will bring few or even no new symbols to the sub-system.
>     - In practice, the stopping criterion is: At most 15 symbols are used for rule extension to avoid semantic redundancy (L209). After that, new symbols are not used for rule extension, leading to a stop of the loop.
>     - With the stopping criterion, the symbol-rule loop satisfies convergence. Take Tab.1 as an example: in rounds 1-5, the newly added symbols count is 5, 5, 3, 3, 1, which shows a convergence trend. Fig.5(a) shows a similar convergence trend. More generally, for the randomly sampled 50 activities, a new round brings an average of 1.9 symbols to the sub-system, verifying the convergence.
> 2. Low sensitivity to initial conditions. The symbol-rule loop should provide a reliable and stable symbolic sub-system over a range of initial conditions.
>     - The sensitivity to initial conditions mainly comes from the asymmetry of rule generation: with 4 as an initial symbol, a rule connecting 4\&10\&6 is generated, while the rule sometimes cannot be generated with 10 as an initial symbol (L214).
>     - However, we find such sensitivity an infrequent case with multiple rules sampling (L215-216). To verify this, we randomly sample 15 different initial symbol sets to generate the sub-system and then measure the average pairwise graph similarity. The average similarity (range: [-1,1]) on 10 randomly sampled activities is 0.91, verifying the low sensitivity to initial conditions.
>
> ### Limitation Discussion
>
> Thanks for your advice. We will add discussions about the bias and stereotypes brought by symbolic computation in the limitation section.

---

> > ### Comment · Reviewer_sPA9 · 2023-08-17
> >
> > Thanks to the authors for their response.  Most of my questions are addressed.  I have a follow up regarding the rule in Figure 6: what you are doing here is exchanging one bias for another, namely exchanging the bias toward detection of background objects as would be present in a pure vision system with a bias toward the stereotypical reasoning I mentioned in the Limitations discussion (that is, the rationality of the rule will bias the output).  In the example you provide, this seems to work well (and I would recommend that you add your answer #4 to the paper for clarification purposes), but there may be cases where this does not improve the results or produces incorrect output (e.g., the presence of a bicycle might bias the output toward "human riding bicycle" even if the person is just standing next to it—see the discussion of false positives in Gkioxari et al "Detecting and recognizing human-object interactions").  This is okay but I would recommend that you be clear about this.  System 1 vs. System 2 reasoning is a cost-benefit tradeoff and simply integrating the two is no guarantee that you nail the right combination for globally better performance.  I will keep my score.

---

> > > ### Author Response · Authors · 2023-08-19
> > >
> > > Thanks for pointing out the potential bias brought by symbolic reasoning. The suggestion is inspiring.
> > >
> > > We will investigate deeper about the difference between System 1 vs. System 2 performance and better integration, which is an interesting and under-explored problem.
> > > We will add more discussion and clarification about Fig.4 visualization results.

---

### Official Review · Reviewer_yZAX · 2023-07-09

**Soundness:** 2 fair
**Presentation:** 3 good
**Contribution:** 3 good
**Rating:** 5
**Confidence:** 2

**Summary:**

This paper studies the use of a symbolic system for understanding human activity. The author argues that a good symbolic system for human activity understanding should possess a diverse range of symbols that accurately identify the human activity, and the rules within the symbolic system should be logically correct. The paper proposes the utilization of LLMs to guarantee these two properties and suggests building a comprehensive pipeline incorporating both LLMs and VLMs. Experiments demonstrate that the proposed system outperforms the baseline model, which solely employs VLMs, on multiple datasets.

**Strengths:**

- The paper is well-written and easy to follow.
- The paper identifies two ideal properties of a good symbolic system: semantic coverage and logical entailment, which is very reasonable.
- Utilizing the commonsense knowledge of LLMs to achieve these two properties is also promising.
The experiments have demonstrated the effectiveness of the model on multiple datasets.

**Weaknesses:**

- Some crucial details of the method are still missing:
    - Regarding symbol initialization, it is not explicitly mentioned how the prompt in Appendix 3.3 (Line 79) is generated. It would be helpful to clarify whether it is manually written or generated using a template, and if so, what is the template.
    - There are two versions of prompts used for rule extension in the main paper (Line 139) and the appendix (Line 84). In the main paper, a single known symbol is used, while in the appendix, all known symbols are used in the prompt. Additionally, in the appendix, the first sentence contains specific information related to the airplane case: "In a picture, **there is an airplane**". How this particular prompt is generated?
    - According to Algorithm 1, symbols that contribute to a rule passing the entailment check should be removed from the candidate symbol list (Line 13). However, in Figure 4, after the 1→6→7→8 rule is generated, there are still rules such as 2→1 and 3→1→9 being generated. It would be useful to provide an explanation for this discrepancy.
- According to Table 1, some generated symbols are more complex than the target activity, such as "wave goodbye to loved ones," and some do not follow the format of "<The person's hands/arms/hip/legs/feet> <verb> <object>", like "airline staff checking the boarding pass". Given the complexity of the generated symbols, VLMs might face challenges in recognizing them in the images. Although experiments have shown that the current pipeline performs better than directly using VLMs on multiple datasets, there is no guarantee that VLMs can easily recognize the generated symbols.
- While the author proposes two ideal properties that should be satisfied, the experiments only focus on the final accuracy of the models on human activity understanding tasks. The accuracy heavily relies on the performance of the VLMs employed. There is no evaluation of how well LLMs ensure these two properties. This makes it unclear what the contribution and bottleneck of the proposed system are. One potential experiment the authors could consider is to have a subset of the dataset and evaluate the pipeline with VLMs replaced by human judgment. But other experiments are also appreciated.

**Questions:**

See my comments in the weaknesses section.

**Limitations:**

The authors have a paragraph discussing the limitations of the proposed method in the last section.

---

> ### Author Rebuttal · Authors · 2023-08-09
>
> We thank the reviewer for the feedback and hope to address the questions and concerns below.
> ### Method Details Explanation
> 1. Symbol Initialization
>     - The prompt is manually designed by us. Special constraints are considered in the prompt to output expected initial prompts. For example, we constrain the answer symbols as hand-related states since they are more common than other symbols (e.g., feet-related states, scenes). The optimal prompt would be found if the prompt can be generated automatically. We leave it to future work.
>
> 2. Rule Extension
>      - The rule extension prompt in the main paper L139 is a brief version, while the prompt in appendix L84 is a more detailed one.
> For the main paper version, "IF [<known symbol>]" refers to listing all the known symbols, which is detailed in the appendix version.
> For human-object interaction activity where an interacted object is known, it is listed in the prompt (e.g., "there is an airplane"), which is omitted in the main paper version. We will add explanations in the final version.
>
> 3. Fig-4 Explanation
>     - We respectfully explain that there is a misunderstanding of Algorithm 1 Line 13. After the 1→6→7→8 rule is generated, the initial symbol 3 is extended into a rule with symbol 1,9 added. According to Line 13, since symbol 1 already exists in $S_c$, it is not pushed into the candidate symbol list to generate the rule 1→6→7→8 repeatedly, and only symbol 9 is added as a new candidate symbol.
> It does not mean that rule 3→1→9 will be removed. Symbol 1 can be reused because it is normal for rules to share some symbols.
>
> ### Symbol Recognition
>
> Though some complex symbols might be difficult to recognize, it does not hinder the main benefit brought by the proposed symbolic system and reasoning pipeline. The analysis is detailed below:
>
> 1. Visual symbols are passed into a checked module, where those with uncertain predictions are filtered out (main paper L183-186, appendix L121-125). Thus, complex symbols which are difficult to recognize are excluded from the reasoning process instead of misleading reasoning.
> 2. After filtering out uncertain symbols, the remaining, easily-recognized symbols are informative enough to facilitate activity reasoning, which is verified via experiments (main paper Tab.2) and visualization examples (main paper Fig. 6).
> 3. Instead of being directly discarded, complex symbols can also be further decomposed into more simple symbols. Then the decomposed simple symbols can facilitate reasoning. For example, "wave goodbye to loved ones" can be decomposed into "arms lift""hand swing" and "eyes look at loved ones". We leave it to future work.
>
> ### Contribution\&Bottleneck Evaluation
>
> Thanks for your advice. We conduct additional evaluations for deeper insight into the results. Currently, the experiment is conducted on a small subset of the HICO test set (120 images for 10 activity classes).
> For each image and each of its ground-truth activities, we list all of the symbols related to the activity in the symbolic system and annotate whether a symbol happens (0/1) in the image via human judgment.
> Conclusions, experiments, and analyses are detailed below. We will scale the experiment and detail it in our final version.
>
> 1. The proposed symbolic system has broader semantic coverage than HAKE.
>     - For quantitative measurement, we count the happening symbols for each image-activity pair under different symbolic systems. The average number is 4.6 for ours and 1.8 for HAKE. Though a rough estimation, the gap in symbol counts indicates that the symbol semantic of HAKE is limited.
> Furthermore, in our symbolic system, different activities have different symbols, further boosting semantic coverage.
> 2. The proposed symbolic system avoids confusion with more rational rules.
>     - We evaluate confusion by counting different image-activity pairs which share the same symbols. For HAKE symbolic system, the confusion problem is severe, with 261 confusion pairs over $C_{120}^2=7140$ pairs (accounting for 3.7\%). For our symbolic system, there are no confusion pairs on the test set because of the presentation ability of symbols and entailment check.
> 3. Reasoning is bottlenecked by System-1 errors (more) and System-2 defects (less).
>     - We investigate the impact of System-1/2 separately. With a perfect System-1, we assume the probabilities of symbols are definitely known as 0/1 instead of $p \in [0,1]$. The ground-truth symbols are annotated as explained above. With a perfect System-2, we assume that the generated rules cover all samples (image-activity pairs). The results are shown below.
>
> | System-1    |  System-2   | mAP        |
> | ----------- | ----------- |----------- |
> | perfect     | perfect     | 100.00     |
> | perfect     | imperfect   | 89.13      |
> | imperfect   | perfect     | 80.07      |
> | imperfect   | imperfect   | 72.59      |
>
> The performance drop [100.00$\rightarrow$89.13 mAP, 80.07$\rightarrow$72.59 mAP] indicates the System-2 defects, i.e., the generated rules have not covered all samples (image-activity pairs). The drop [100.00 $\rightarrow$ 80.07 mAP, 89.13 $\rightarrow$72.59 mAP] indicates the System-1 errors, i.e., symbol probabilities cannot be predicted accurately. On this test set, System-1 errors have more negative impacts on the final performance.

---

> > ### Comment · Reviewer_yZAX · 2023-08-15
> >
> > Thanks for the clarification and the additional experiment results. From the response, the prompts for symbol initialization and rule extension are manually designed for different tasks and datasets (e.g. the symbols are constrained to hand-related states for HOI recognition). I have a number of questions regarding these prompts:
> > 1. What are the prompts for every dataset/task? It is very important to include the original prompts in the paper for the purpose of reproducibility and clarity.
> > 2. How are these prompts determined?
> > 3. Are experiment results sensitive to the design choice of prompts?
> >
> > I am a little concerned about how much extra effort we need to tune the prompts to have a good performance. I understand that it is hard to have a systematic way of evaluating the effect of prompts. And it might be hard to generate prompts that generalize across different domains. But it is valuable to include discussions and details for a paper that heavily leverages LLMs, because LLMs are known to be sensitive to prompt design.

---

> > > ### Author Response · Authors · 2023-08-19
> > >
> > > We thank the reviewer for the feedback. Questions are answered below.
> > > ###  What are the prompts for every dataset/task?
> > > The prompts for the proposed dataset/task related to activity understanding are basically similar.
> > > The prompt differences are small modifications necessary to adapt different dataset/task settings:
> > >
> > > - When an interacted object is known, it is listed in the rule extension prompt for emphasis.
> > > - For HAKE-PaSta, the symbol initialization is omitted. (suppl L147-150)
> > >
> > > Also, we respectfully point out it is a misunderstanding that "hand-related states" is "tailored for HOI", according to our response above: "...constrain the answer symbols as hand-related states since they are more common than other symbols...".
> > > We attach the original prompt in Appendix 1 in the following response for further investigation.
> > >
> > > ### How are these prompts determined?
> > > The prompts are determined via a trial-and-error process by human experts on randomly sampled 10 activities.
> > > Human experts have read the image samples of these activities, and thus have expert knowledge about what symbols/rules are more suitable for visual reasoning.
> > > Each round in the trial-and-error process is as follows:
> > >
> > > - one expert writes a prompt
> > > - an LLM generates answers by the prompt
> > > - all experts judge whether the answer is suitable
> > > - all experts modify the prompt based on the discovered problems
> > >
> > > An example of the trial-and-error process is detailed in Appendix 2 in the following response.
> > > We admit the current process needs improvement: the human experts are limited to the authors, and the prompts are verified on a few activities.
> > > However, it is a performance-cost trade-off: as the reviewer points out, it takes extra effort to tune the prompts to have a good performance.
> > >
> > > ### Are experiment results sensitive to the design choice of prompts?
> > > Experiment results are sensitive to prompts with major differences. Using the older version prompt during the trial-and-error process brings difficulty to correct and automated symbols generation.
> > >
> > > Experiment results are less sensitive to prompts with minor differences, i.e., slightly paraphrasing the prompts. We conduct a brief experiment:
> > >
> > > - get another 2 paraphrased symbol init prompts and another 2 paraphrased rule extension prompts
> > > - use these $3*3=9$ prompts to generate 9 sub-systems
> > > - measure the average pairwise graph similarity
> > > - the average similarity (range: [-1,1]) on 10 randomly sampled activities is 0.83

---

> > > > ### Author Response · Authors · 2023-08-19
> > > >
> > > > ### Appendix1
> > > > For an activity with an interacted object (e.g., board an airplane) in HICO, HAKE-Verb, Stanford40:
> > > >
> > > > - (Symbol Initialization) In a picture, a person is boarding an airplane. What are his hands doing? Answer with 5 concise, highly-related phrases. The format is "<hands> <verb> <object>".
> > > > - (Rule Extension) In a picture, there is an airplane. IF [The person is holding a boarding pass...] AND [condition] THEN [The person is boarding the airplane.] [condition] is one concise phrase. The format is "<The person’s hands/arms/hip/legs/feet> <verb> <object>". What is [condition]?
> > > >
> > > > For an activity with no interacted object (e.g., cook) in HICO, HAKE-Verb, Stanford40:
> > > >
> > > > - (Symbol Initialization)
> > > > In a picture, a person is cooking. What are his hands
> > > > doing? Answer with 5 concise, highly-related phrases. The format is
> > > > "<hands> <verb> <object>".
> > > > - (Rule Extension)
> > > > In a picture, IF [The person is holding a slice...] AND [condition] THEN [The person is cooking.] [condition] is one concise phrase. The format is
> > > > "<The person’s hands/arms/hip/legs/feet> <verb> <object>". What is
> > > > [condition]?
> > > >
> > > > For a PaSta (e.g., hold something in both hands) with a known activity (e.g., wash an airplane) in HAKE-PaSta:
> > > >
> > > > - (Symbol Initialization) None
> > > > - (Rule Extension) In a picture, IF [The person is washing an airplane...] AND [condition] THEN [The person is holding something in both hands.] [condition] is one concise phrase. The format is "<The person’s hands/arms/hip/legs/feet> <verb> <object>". What is [condition]?
> > > >
> > > >
> > > > ### Appendix2
> > > > Trial-and-error process of the prompts for symbol initialization
> > > >
> > > > Round1:
> > > >
> > > > - Prompt: In a picture, a person is boarding an airplane. What are useful visual features to distinguish his activity?
> > > > - Answer: Useful visual features...could include: 1. Luggage: Look for the presence of suitcases, backpacks, or other types of bags that suggest the person is preparing to travel...
> > > > - Comment: Object-centric, difficult to extract key information about human activities.
> > > >
> > > > Round2:
> > > >
> > > > - Prompt: In a picture, a person is boarding an airplane. What is he doing?
> > > > - Answer: The person in the picture is boarding an airplane. Boarding refers to the process of entering the aircraft before the flight begins. It typically involves presenting a valid boarding pass, showing identification, and walking through the designated gate or bridge to reach the plane.
> > > > - Comment: Verbose.
> > > >
> > > > Round3:
> > > >
> > > > - Prompt: In a picture, a person is boarding an airplane. What is he doing?
> > > > Answer with 5 concise, highly-related pharses. The format is "<human> <verb> <object>".
> > > > - Answer: Person boarding airplane.
> > > > - Comment: The answer repeat the question.
> > > >
> > > > Round4:
> > > >
> > > > - Prompt: In a picture, a person is boarding an airplane. What are his hands doing?
> > > > Answer with 5 concise, highly-related pharses. The format is "<hands> <verb> <object>".
> > > > - Comment: Satisfactory.

---

> > > > > ### Comment · Reviewer_yZAX · 2023-08-21
> > > > >
> > > > > Thanks authors for the response. I think the current system provides a reasonable way to leverage LLMs and VLMs to do visual activity understanding. But according to the experiment results, the prompting method can critically affect the performance, limiting the robustness and generality of the system. I will keep my current rating.

---

> > > > > > ### Author Response · Authors · 2023-08-21
> > > > > >
> > > > > > We thank the reviewer for the follow-up feedback. The suggestion provides a deeper analysis of our method.
> > > > > >
> > > > > > Nevertheless, we hold a slightly different view about the prompting method:
> > > > > >
> > > > > > - Prompting affect the performance: We leverage a trial-and-error process to find the proper prompt. Minor modification on the final prompt has a relatively low impact on the topology of the symbolic system.
> > > > > > - Robustness: Modified prompts lead to different results. This is not necessarily a limitation of robustness but can be a way to increase diversity, i.e., various reasonable symbols/rules can be generated via prompting without much extra effort.
> > > > > > - Generality: The trial-and-error process can be reused by other tasks.
> > > > > >
> > > > > > We mainly focus on the insight of two properties in our initial submission and did not preferentially consider the prompt design, which is an alternative implementation. We will add more analysis and discussion about the impact of prompting. Thanks again for your suggestion!

---

### Official Review · Reviewer_zui3 · 2023-07-14

**Soundness:** 3 good
**Presentation:** 3 good
**Contribution:** 2 fair
**Rating:** 6
**Confidence:** 3

**Summary:**

In this paper, a symbolic-System-2 submodule is integrated with a System-1 module to perform human activity recognition. The work follows the intuitive idea, where a visual prediction module extracts visual symbols and a symbolic reasoning module performs logical reasoning to recognize activities. For the visual symbol extraction module, the work applies a general VLM, to predict the truth value of a predicate in a logic clause. The logical system then performs fuzzy logic to predict the final outcome of the rule, ie, conclusion. The logical system is constructed from LLM, without expensive human annotation in existing works. In experiments, it is shown that the model is better than existing baselines.

**Strengths:**

+ The work leverages and extends the intuitive idea of System-1 and System-2 combination. The integration is important compared to System-1 only pure predictive framework and it's really encouraging to see more and more works integrating these two systems in typical vision tasks.
+ Compared to existing works, the method leverages LLM for knowledge-base extraction. While the cost saving is from LLM, rather than some nice problem structure design, I do appreciate the efforts to construct formal logic base from natural language systems and make it work.


**Weaknesses:**

I'm not an expert in logical system design so I will refrain from commenting on the system design part on logics. However, I do see some general problems regarding this work.

For one, designing logical system specifically for a task risks overfitting. While the performance is improved on a specific task, I do note that the slight increase may not be worth the generality from models like CLIP. Besides, the logic system is pretty complicated. And I do not think the system, or the system construction method could be easily adapted for other tasks. On the other hand, such a system is also only demonstrated in a single task. I doubt if the method could be more general.

I'm also interested in seeing more detailed ablation study. In particular, whether it is the System-1 part of the System-2 part that more adversely impacts performance. Would it be possible to assume perfect System-1 and test the System-2 performance or vise versa? Besides, while in general the idea should work. There are many subtleties in telling the truth value of statements, eg, the leg is ``close'' to something. How should such subtleties be addressed?

Those being said, I admit the inherent conflict between logical system that are delicate, explainable, robust, and generalizable and deep learning system that are general but fragile. I do not see these points as weaknesses but rather interested in hearing how the authors see them.

For my final rating, I'd like to sync with fellow reviewers.

**Questions:**

See above.

**Limitations:**

Yes.

---

> ### Author Rebuttal · Authors · 2023-08-09
>
> We thank the reviewer for the feedback and hope to address the questions and concerns below.
> ## Generality
> The method could be more general. We choose activity understanding as a good and important initial test bed because it is a difficult task with complex visual patterns and compositional nature. Based on the contribution of this paper, we step towards solving the integration of System-1/2 for general visual understanding.  The proposed symbolic system can be generalized to various tasks (e.g., classification, visual commonsense) with a similar formulation. Some examples are shown below:
>
> 1. Object classification
>     - For the object "crocodile" in CIFAR-100 [1] dataset, one example rule is: long, slightly curved body $\wedge$ four short legs $\wedge$ ... $\wedge$ a long, muscular tail $\rightarrow$ crocodile
>
> 2. Visual commonsense
>     - For one example QA in VCR [2] dataset (shown in attached pdf Fig.1), one example rule is: person on the right shows affectionate gestures $\wedge$ person on the left engages in conversation $\wedge$ ... $\wedge$ The environment is a restaurant $\rightarrow$ b
>
> Under this formulation, semantic coverage and logical entailment remain two important properties under a more general setting, and we can reason with visual inputs via a similar pipeline.
> We conduct experiments with two zero-shot benchmarks:
>
> 1. CIFAR-100 [1] test set.
>
> | Method           | CIFAR-100 Acc (\%) |
> | -----------      |  -----------        |
> | CLIP             | 77.50               |
> | CLIP+Reasoning   | 78.31               |
>
>
> 2. VCR [2] val set. BLIP2 is adopted as a baseline as it is a VQA-style task. VCR val set has 26534 image-question pairs, where we select a subset of 557 pairs for the test. The selected pairs are highly related to human activities (instead of role, mental, .etc) and only involve *one* person's activity.
>
> | Method            | VCR Acc (\%)|
> | -----------       | ----------- |
> | BLIP2             | 46.32        |
> | BLIP2+Reasoning   | 47.08        |
>
> We can see that the proposed pipeline can be applied to general visual tasks and facilitate reasoning. In this paper, we focus on method design instead of task generalization.
> We will add these explanations in the final version and analyze more general settings in future work.
>
> [1] Krizhevsky, Alex, and Geoffrey Hinton. "Learning multiple layers of features from tiny images." (2009): 7.
>
> [2] Zellers, Rowan, et al. "From recognition to cognition: Visual commonsense reasoning." Proceedings of the IEEE/CVF conference on computer vision and pattern recognition. 2019.
>
> ## System-1/2 Ablation
>
> Thanks for your advice. We conduct a detailed ablation study on a small subset of the HICO test set (120 images for 10 activity classes). We investigate the impact of System-1/2 separately.
> With a perfect System-1, we assume the probabilities of symbols are definitely known as 0/1 instead of $p \in [0,1]$. With a perfect System-2, we assume that the generated rules cover all samples (image-activity pairs). To construct a perfect System-1, for each image and each of its ground-truth activities, we list all of the symbols related to the activity in the symbolic system and annotate whether a symbol happens (0/1) in the image via human judgment. The results are shown below.
>
> | System-1    |  System-2   | mAP        |
> | ----------- | ----------- |----------- |
> | perfect     | perfect     | 100.00     |
> | perfect     | imperfect   | 89.13      |
> | imperfect   | perfect     | 80.07      |
> | imperfect   | imperfect   | 72.59      |
>
> The performance drop [100.00$\rightarrow$89.13 mAP, 80.07$\rightarrow$72.59 mAP] indicates the System-2 defects, i.e., the generated rules have not covered all samples (image-activity pairs). The drop [100.00 $\rightarrow$ 80.07 mAP, 89.13 $\rightarrow$72.59 mAP] indicates the System-1 errors, i.e., symbol probabilities cannot be predicted accurately. On this test set, System-1 errors have more negative impacts on the final performance.
> We will scale the experiment and detail it in our final version.
>
> ## Symbol Recognition Subtleties
>
> Though some complex symbols (e.g., the leg is "close" to something) might be difficult to recognize, such subtleties do not hinder the main benefit brought by the proposed symbolic system and reasoning pipeline. The analysis is detailed below:
>
> 1. Visual symbols are passed into a checked module, where those with uncertain predictions are filtered out (main paper L183-186, appendix L121-125). Thus, complex symbols which are difficult to recognize are excluded from the reasoning process instead of misleading reasoning.
> 2. After filtering out uncertain symbols, the remaining, easily-recognized symbols are informative enough to facilitate activity reasoning, which is verified via experiments (main paper Tab. 2) and visualization examples (main paper Fig. 6).
> 3. Instead of being directly discarded, some complex symbols can also be further decomposed into more simple symbols. Then the decomposed simple symbols can facilitate reasoning. For example, "wave goodbye to loved ones" (an example by Reviewer yZAX) can be decomposed into "arms lift", "hand swing", and "eyes look at loved ones". We leave it to future work.

---

> > ### Comment · Reviewer_zui3 · 2023-08-15
> >
> > Thanks for the clarification. The response is satisfactory. I will keep my accept vote unless other reviewers point out critical issues I missed.

---

### Official Review · Reviewer_K3QA · 2023-07-25

**Soundness:** 2 fair
**Presentation:** 2 fair
**Contribution:** 2 fair
**Rating:** 3
**Confidence:** 3

**Summary:**

The paper addresses the problem of visual activity understanding and proposes a way to leverage large visual and language models to i) propose symbols from an image, ii) derive new symbols via logic entailment and perform classification. The results show that the augmented system with symbolic reasoning capabilities performs slightly better than the equivalent large models without the reasoning system.

**Strengths:**

The paper proposes an interesting way to leverage large visual and language models to derive a set of symbols relevant to a particular picture and the action being executed in it. Constructing an ontology based on the compiled knowledge represented in an LLM seems to be a powerful approach to enable task specific higher level reasoning.

**Weaknesses:**

One of the key improvements over other related work is claimed to be the broad semantic coverage of the derived symbols and the generated rules. However, given the statistical nature of the used large models it is not clear how accurate these symbols and rules actually are. A deeper analysis of the generated ontology is definitely needed in order to assess the applicability of the proposed method.

**Questions:**

- How is the set of the initial symbols selected? Isn't this initial set generation prone to the same issues as some of the related works?
- Given the form of each rule having multiple premises and a single conclusion, why is the hyper-graph structure needed? Wouldn't a simple graph with multiple directed edge be enough?
-

**Limitations:**

The core of the proposed work is based on utilising pre-trained large models which require vast amounts of data and computational resources and so the proposed method is as efficient as described only by assuming access to such large models.

---

> ### Author Rebuttal · Authors · 2023-08-09
>
> We thank the reviewer for the feedback. We will add more discussions about limitations brought by using pre-trained large models. We hope to address the questions and concerns below.
>
> ## Detailed Evaluation\&Analysis
>
> We respectfully explain that using LLM is the optimal method within our consideration, which balances abundant knowledge and low manual cost. More reliable knowledge than LLM should be extracted from human annotations, but complete human evaluation of generated symbols \& rules is quite expensive.
> Therefore, for deeper analysis, we currently focus on a small but representative subset. On this test subset, we verify whether the proposed symbolic system indeed brings advantages (e.g., broader semantic coverage, and more rational rules to avoid confusion). Also, we analyze the performance bottleneck of System-1 errors and System-2 defects. Furthermore, we analyze and evaluate the robustness of the generated ontology.
> Details are shown below. We will scale the experiment and detail it in our final version.
>
> ### The test set
> We select a small subset of the HICO test set (120 images for 10 activity classes). The 10 activity classes are: "human board/wash an airplane", "human park/repair a bicycle", "human feed/race a horse", "human break/sign a baseball bat", and "human buy/wash an orange". These activities have relatively complex visual patterns and cover different interacted objects.
> For each image and each of its ground-truth activities, we list all of the symbols related to the activity in the symbolic system and annotate whether a symbol happens (0/1) in the image via human judgment.
>
> ### Symbols\&Rules Evaluation
> The proposed symbolic system has broader semantic coverage than HAKE. For quantitative measurement, we count the happening symbols for each image-activity pair under different symbolic systems. The average number is 4.6 for ours and 1.8 for HAKE. Though a rough estimation, the gap in symbol counts indicates that the symbol semantic of HAKE is limited. Furthermore, in our symbolic system, different activities have different symbols, further boosting semantic coverage.
>
> The proposed symbolic system avoids confusion with more rational rules.
> We evaluate confusion by counting different image-activity pairs which share the same symbols. For HAKE symbolic system, the confusion problem is severe, with 261 confusion pairs over $C_{120}^2=7140$ pairs (3.7\%). For our symbolic system, there are no confusion pairs on the test set because of the presentation ability of symbols and entailment check.
>
> ### Bottleneck Analysis
> We investigate the impact of System-1/2 separately as below.
> With a perfect System-1, we assume the probabilities of symbols are definitely known as 0/1 instead of $p \in [0,1]$. The ground-truth symbols are annotated as explained above.
> With a perfect System-2, we assume that the generated rules cover all samples (image-activity pairs).
>
> | System-1    |  System-2   | mAP        |
> | ----------- | ----------- |----------- |
> | perfect     | perfect     | 100.00     |
> | perfect     | imperfect   | 89.13      |
> | imperfect   | perfect     | 80.07      |
> | imperfect   | imperfect   | 72.59      |
>
> The performance drop [100.00$\rightarrow$89.13 mAP, 80.07$\rightarrow$72.59 mAP] indicates the System-2 defects, i.e., the generated rules have not covered all samples (image-activity pairs). The drop [100.00 $\rightarrow$ 80.07 mAP, 89.13 $\rightarrow$72.59 mAP] indicates the System-1 errors, i.e., symbol probabilities cannot be predicted accurately. On this test set, System-1 errors have more negative impacts on the final performance.
>
> ### Robustness Evaluation
> Convergence. The symbol-rule loop should find a symbolic sub-system that "converges". In other words, after a certain stopping criterion, a new round will bring few or even no new symbols to the sub-system.
> In practice, the stopping criterion is: At most 15 symbols are used for rule extension to avoid semantic redundancy (L209).
> With this stopping criterion, the symbol-rule loop satisfies convergence. For the randomly sampled 50 activities, a new round brings an average of 1.9 symbols to the sub-system, verifying the convergence.
>
> Low sensitivity to initial conditions. The symbol-rule loop should provide a reliable and stable symbolic sub-system over a range of initial conditions.
> We randomly sample 15 different initial symbol sets to generate the sub-system and then measure the average pairwise graph similarity. The average similarity (range: [-1,1]) on 10 randomly sampled activities is 0.91, verifying the low sensitivity to initial conditions.
>
> ## Method Details Explanation
> ### Initial Symbols Selection
>
> The initial symbols are five hand-related states related to the activity, which are generated by prompting an LLM (L147, appendix L78-81).
>
> The initial symbols are simply used as triggers for rule generation, while the rule extension is the main component to extract LLM knowledge and guarantee broad semantic coverage. Fig. 4 and Tab. 1 show an example where initial symbols can expand to a satisfying symbolic sub-system via symbol-rule loop and entailment check. Furthermore, the symbol-rule loop has low sensitivity to initial symbols (see above). Thus, the initial symbols have a relatively low impact on a symbolic sub-system.
>
> ### Hyper-graph Structure
> In the symbolic system, there are two "multiple" relationships: each activity has multiple rules, and each rule has multiple premises. We find hyper-graph a proper structure to clarify it, where one rule is a hyperedge with connecting multiple vertices (premises). Maybe other alternative structures can also be used to clarify the system.

---

> ### Comment · Area_Chair_Gdfa · 2023-08-18
>
> Dear Reviewer K3QA,
>
> We are nearing the end of the discussion period with authors.
>
> The authors have responded in detail to your review, so pls minimally read and acknowledge their rebuttal, and state which (if any) issues you still do not find to be satisfactorily addressed.
>
> You should do so as soon as possible.
>
> Thanks,
> AC

---

### Official Review · Reviewer_SHNq · 2023-08-01

**Soundness:** 3 good
**Presentation:** 3 good
**Contribution:** 2 fair
**Rating:** 5
**Confidence:** 3

**Summary:**

The paper proposes to increase coverage and generalization of logic-based methods to model 'system-2' or deliberative reasoning for the task of human activity prediction in images. Specifically, the authors incorporate an LLM (GPT3.5) to obtain a larger number of symbols and inference rules related to human activity prediction and further also utilize an LLM to check logical entailment of generated rules (based on which noisy rules are filtered). Their proposed symbolic system operates upon a VLM which is used for symbol extraction from the image, based on which rules are activated with a fuzzy-logic computation (to obtain a probability score for the inference chain).

Authors show that adding their method on top of two VLMS (CLIP and BLIP2), enhances their performance on prominent activity recognition benchmarks and also show visualizations of activated rules and predictions besides ablation studies of their method.

**Strengths:**

1. The usage of LLMs to broaden semantic coverage of symbolic/logic-based methods by discovering relevant premises and rules and also for checking/approximating logical entailment is relatively novel.

2. The methodology and formalism of their approach is well described, with appropriate examples and figures/tables to help reader understand the approach.

3. Results on multiple prominent benchmarks (including HAKE, HICO and Stanford-40) for human activity prediction show that the method can benefit both zero-shot (for BLIP2 and CLIP) and finetuned (for CLIP) performances of prominent VLMs. Ablation results also highlight contributions of individual method components/variations.

**Weaknesses:**

1. The symbolic system's application draws on using a VLM multiple times to extract the probability of individual symbols. It thus seems that the computation will linearly scale with number of symbols, which can be a significant drawback due to the incurred computation cost.  (VLMs such as BLIP2 by themselves have very large computation cost, hence even increasing that 100x for 100 symbols is a very significant additional compute). Further, extending this to further domains beyond human-activity recognition would require many more symbols, and ultimately possibly not be scalable.

2. Evaluation baselines only consider results of the backbone VLMs currently. There are multiple recent works in related direction of human-object interaction and activity prediction in zero- or few-shot scenarios which have not been mentioned or used as baselines (e.g. Weakly-Supervised HOI detection (ICLR 2023); RLIP (NeurIPS 2022)). Currently only RelVIT is mentioned in supplemental. Authors should either compare with such methods or mention why these recent methods are not used/suitable for comparison.

3. Further, since LLMs are employed for much of the 'system-2' reasoning, authors should consider reporting results of alternative methods using VLMs + LLMs directly without any logical system (e.g. VLM generates image caption, based on which LLM predicts possible activity when prompted for each activity in the label space -- similar to PICa in  "An Empirical Study of GPT-3 for Few-Shot Knowledge-Based VQA" AAAI 2022). These results could better highlight the contribution of the proposed logical system and that performance increments are not due to just LLM-VLM combination.

Less important:
1. Failure cases could be illustrated for a small sample of the questions to better highlight which component is currently leading to errors (e.g. is it the VLM not producing accurate symbols or is it a noisy rule, etc)

**Questions:**

Please see weaknesses.

My primary concerns (impacting rating) are regarding:
1. Does compute scale linearly with number of symbols and would this be a major limitation in practice? If so, can this can be addressed through alternative symbol grounding/identification methods?
2. The relatively limited evaluation compared to other relevant works for human activity prediction. How does the method compare to related baselines in human-object interaction learning such as ICompass (Interaction Compass: Multi-Label Zero-Shot Learning of Human-Object Interactions via Spatial Relations (ICCV 2021) and RLIP (NeurIPS 2022)) or alternatively multimodal LLM methods (e.g. PICa in "An Empirical Study of GPT-3 for Few-Shot Knowledge-Based VQA" AAAI 2022)?

**Limitations:**

Potential negative societal impact is mentioned to be minimal.

---

> ### Author Rebuttal · Authors · 2023-08-09
>
> We thank the reviewer for the feedback and hope to address the questions and concerns below.
> ## Computation Scalability
>
> Symbol predictions will increase the computational cost as a trade-off for explainability and generalization. However, with the proper method design shown below, the computational cost can be largely reduced and will not be a major limitation.
>
> 1. Not all symbols need to be predicted via pruning. Symbols can be organized as a tree with hierarchical prediction.
> For the Fig.6 example, $m_1$...$m_{10}$ can be organized as:
>     - $m_a$: Interact with a seller:
>         - $m_1$:Talk with seller
>         - $m_3$: Seller hand over orange
>         - $m_{10}$: Give money to seller
>     - $m_b$: Interact with the oranges:
>         - $m_2$: Reach for an orange
>     - $m_c$: Interact with a container:
>         - $m_5$: Place orange in a bag
>         - $m_6$: Pick orange from a basket
>         - $m_7$: Hold a bag of oranges
>         - $m_9$: Seller put the orange in bag
>     - $m_d$: Payment Process:
>         - $m_4$: Stand in front of fruit stand
>         - $m_8$: Reach for a wallet
>
> The father symbols (e.g., $m_a$)
> can be summarized based on the son symbols (e.g., $m_1$) via simple semantic extraction (e.g., LLM prompt).
> On the symbol tree, the probability of a son symbol is no higher than its father symbol.
> When the probability of a father symbol is lower than a threshold (e.g., in image III, $m_a$ probability less than 0.1), its son symbols ($m_1$, $m_3$, $m_{10}$) will be assigned the threshold directly without extra computation.
>
> Thus, the computational cost is saved via the hierarchical structure of symbols. For example, image III only needs 5 instead of 10 calculations: Interact with a seller(x), Interact with the oranges, Reach for an orange, Interact with a container(x), and Payment Process(x). The computation save will be more evident with the number of symbols scales.
>
> 2. Activity prediction can be determined with only part of known symbols, which benefits from the fuzzy-logic calculation with the simple minimum/maximum function.
> Take the image I in Fig.6 as an example. When the probabilities of $m_1=0.92$, $m_2=0.14$, $m_3=0.94$, $m_4=0.57$ are known, we have $p(r_1)=0.14$, $p(r_2)=0.92$, $p(r_3),p(r_4),p(r_5)\leq0.57$, thus determining the activity prediction $p(c)=0.92$ with only 4 known symbols.
> 3. Different activities have different complexity and computational cost.
>     - For those simple activities (e.g., "hold an apple"), they have few rules or can even be predicted without reasoning. Thus, the computational cost is controllable.
>     - Our reasoning is especially beneficial to those complex activities (e.g., "board an airplane") with various patterns and insufficient training samples. Thus, the computational cost proposed symbolic system is a trade-off to compensate for the data insufficiency and solve complex activities.
>
> ## VLMs+LLMs Ablation
> To highlight the contribution of the proposed logical system, we conduct experiments where VLMs and LLMs are combined in different ways. Results on zero-shot Stanford-40 are shown below.
>
> 1. LLM only provides implicit reasoning for VLM (BLIP2): 91.85 mAP
> 2. LLM is equipped with rules, and the rules are prompted in the label space
>     - Prompt LLM in the VLM. The text prompt of BLIP2 contains not only the target activity but also its rules generated from our symbolic system: 91.97 mAP
>     - Prompt an external LLM like PICa. BLIP2 generates image captions, and an LLM make activity prediction based on the caption and the rule prompts: 73.18 mAP
> 3. LLM is equipped with rules and explicit reasoning (ours): 92.59 mAP
>
> We can find that the proposed explicit reasoning is superior to using rules as LLM prompts because of the precise knowledge representation. When prompting rules in LLM's label space, inserting the prompt in a VLM is superior. Using an external LLM fails possibly because the image caption fails to capture the accurate semantics for LLM to judge. However, there are more designs left to explore, e.g., in-context learning in PICa.
>
> ## HOI Learning Baselines
>
> We respectfully explain that we focus the reasoning methodology on *image-level* activity understanding benchmarks. Thus, only methods (e.g., RelViT) on HOI recognition benchmark HICO are listed as baselines, and HOI detection methods (e.g., Weakly-HOI, RLIP) on HICO-DET are not listed. Since HOI detection requires additional designs for detecting human-object pairs, we leave it to future work.
> Instead, we use several other image-level activity understanding benchmarks, e.g., Stanford40 (action recognition), and HAKE-PaSta (conditional PaSta Q-A). Our experiment settings are more similar to RelViT (focus on reasoning on HICO, GQA dataset) than to HOI detection methods (focus on HICO-DET, VCOCO dataset).
>
> Additionally,  ICompass shows zero-shot HOI recognition performance in Tab. 1, where some HOI categories are seen and others are unseen. Our method focuses on reasoning while ICompass focuses on action/object score composition. Thus, we currently use it simply as a baseline and will add it in the updated version.
>
>
> ## Failure Cases Analysis
>
> We provide two failure cases (false negative) for the activity "human board an airplane" in the attached pdf Fig.2.
> In image I, the failure is caused by the prediction error of symbol probabilities (which we name "System-1 error"). Some symbols are predicted as low probabilities by mistake, e.g., "walk towards the boarding gate". The symbol prediction error is possibly caused by tiny human bounding boxes.
> In image II, the failure emerges because the generated rules do not cover the situation in the image, i.e., "System-2 defects". For example, the image contains the symbol "climb up the airplane edge" which is not in the symbolic system.
>
> We also make quantitative analysis on System-1 errors and System-2 defects, which is detailed in the response for Reviewer zui3 "System-1/2 Ablation".

---

> > ### Comment · Reviewer_SHNq · 2023-08-15
> >
> > Thanks for the detailed response and clarifications.
> >
> > Regarding point 1 on computation scalability -- I agree that a hierarchical structure as described should reduce computational cost. However, it is unclear whether this is currently integrated in the proposed framework and reported results? Or is it a proposed extension for scalability?
> > In relation, a quantitative comparison (e.g. number of operations/FLOPs) could be beneficial and provide more clarity on this potential issue.
> >
> > It is also unclear to me whether the probabilities of symbols are to be retrieved for each potential activity class? E.g in Fig. 6 Image I-IV, currently relevant symbols are shown for 'buy orange' class; would the probability retrieval also be done for other relevant symbols of other possible classes (such as 'visit store', etc) for each image? (in this case, it seems the number of total symbols would be larger than 5 and scale with number of classes)
> >
> > This is my remaining major concern. My other concerns have been addressed adequately. The VLMs + LLMs ablation results could be added in the main paper to highlight the symbolic system's benefits. Similarly, quantitative analysis on System-1 errors and System-2 defects could be added to main paper.

---

> > > ### Author Response · Authors · 2023-08-19
> > >
> > > We thank the reviewer for the feedback. We will update additional results and analysis. Questions are answered below.
> > >
> > > ### The hierarchical structure
> > > The hierarchical structure is a proposed extension for scalability.
> > > We mainly focus on the effectiveness and implementation of our insight in our initial submission and did not preferentially consider the efficiency.
> > > Nevertheless, it is easily integrated into the proposed framework, with a new hierarchical connection applied to the original symbolic system.
> > >
> > > ### Are symbols retrieved for each potential activity class?
> > > The total number of symbols to measure will increase with the number of potential activity classes, but not scale linearly.
> > >
> > > - It is unavoidable to measure probabilities of more than one activity class for activity understanding tasks. This is because activity recognition is a multi-label classification rather than a multi-class classification: A person typically performs multi-action simultaneously, e.g., standing while eating. Thus, the increase in symbol measurement comes from the inherent characteristic of activity tasks.
> > > - However, the symbols are reused for different activity classes without repetitive computation. For example, "buy orange" and "visit store" shares the symbols: ($m_a$) Interact with a seller, ($m_d$) Payment process, ($m_1$) Talk with seller, ($m_{10}$) Give money to seller, ($m_8$) Reach for a wallet. It is also illustrated in the main paper Fig.2(a): the red and green activities share some symbols (gray), e.g., second from right, third from left.
> > >
> > > ### Quantitative comparison
> > > To improve clarity, we report the num-of-operations average per image on HAKE-Verb. An "operation" is defined as predicting a symbol or an activity class directly.
> > >
> > > |  reasoning     |  num-of-operations    |
> > > | -----------    | -----------           |
> > > | w/o            |  18                   |
> > > | w/             |  39                   |
> > >
> > > Without reasoning, activity classes are predicted directly, and num-of-operations vary from images due to different ground-truth object labels which verb prediction conditions upon (suppl L138).
> > > With reasoning, reused symbols are predicted via a hierarchical structure.
> > >
> > > We provide a more detailed analysis of a small case. There are 38 images known to contain oranges, where verbs (buy/cut/eat/hold/inspect/peel/pick/squeeze/wash) are to be predicted.
> > > Without reasoning, num-of-operations=9 (same for each image).
> > > With reasoning:
> > >
> > > |  reuse     |  hierarchical     |   num-of-operations     | remark |
> > > | ----------- | ----------- |----------- |----------- |
> > > | w/o     | w/o     | 71     | same for each image |
> > > | w/     | w/o   | 31      |  same for each image|
> > > | w/     | w/     | 23     | average over 38 images|

---

> > > > ### Comment · Reviewer_SHNq · 2023-08-21
> > > >
> > > > Thanks for the further details and clarifications. Based on the number of operations, I believe the additional computation is relatively significant (as mentioned 18 operations w/o reasoning to 39 with reasoning, which is more than double). Further, when applying method to new domains, additional symbols and rules would need to be extracted/identified which is dependent on underlying VLMs and LLMs and is more complex than direct finetuning / few-shot application of VLMs. While these may be addressable through future work, I encourage authors to explicitly highlight these points on computation scalability and applicability to other domains (with dependence on VLMs/LLMs) in the limitations section of their current work.
> > > >
> > > > I have raised my score to 5 given the work's novelty and performance benefits.

---

> > > > > ### Author Response · Authors · 2023-08-21
> > > > >
> > > > > We thank the reviewer for the insightful suggestion!
> > > > >
> > > > > We will discuss computation scalability and applicability to other domains in the limitation section, and explore these important issues in future work.

---

### Official Review · Reviewer_sNaH · 2023-08-02

**Soundness:** 3 good
**Presentation:** 3 good
**Contribution:** 2 fair
**Rating:** 4
**Confidence:** 4

**Summary:**

The paper proposes a novel method for visual activity understanding using an enriched symbolic system. The authors focus on enhancing the integration of intuitive (System-1) and logical (System-2) reasoning processes in AI systems. They argue that the current System-2 reasoning methods suffer from limitations such as a dearth of symbols and limited, ambiguous rules, inhibiting the system's ability to capture the complex patterns of activities and limiting its explainability, generalization, and data efficiency.

In response to these shortcomings, the authors introduce a new symbolic system enriched with a broader set of symbols and more rational rules. This system, underpinned by commonsense knowledge, aims to foster a more comprehensive understanding of human activities visually. To support their work, they leverage the advancements in LLMs and techniques like the symbol-rule loop and entailment check.


**Strengths:**

The concept of developing a new symbolic system that combines aspects of intuitive (System-1) and logical (System-2) reasoning is indeed a creative approach. However, the idea of incorporating commonsense knowledge into AI reasoning is not entirely novel, as there have been previous works exploring similar notions.

The authors put forward a detailed methodology and apply recent advancements such as Large Language Models (LLMs) in their approach.

While the paper is mostly well-written, certain parts, especially the description of the symbolic system and its instantiation, could benefit from more elaboration or clearer explanations.

Although the paper targets a significant area in AI - the integration of System-2 reasoning into visual activity understanding - it's unclear how much practical impact this specific method would have on the broader field. The lack of comparison with other methods, especially in various real-world applications, makes it difficult to evaluate the actual significance of their work.


**Weaknesses:**

The paper's evaluation approach, while it encompasses multiple datasets, does fall short in providing a comprehensive set of comparisons with current state-of-the-art methods. The selected baselines do not reflect the full breadth of existing approaches, which undermines the claim of superiority of the proposed method. It would be valuable to include comparisons with more recent vision-language models, such as LLaVA, MiniGPT-4, and LLaMA Adapter, to provide a more meaningful assessment of the proposed system's performance.

The paper does spend a substantial portion of its content introducing the method background. Shifting sections 3.1-3.2 to the appendix could help maintain focus on the main contribution of the paper, while still providing the necessary context for interested readers.

The explanation of the core methodology is another area that requires improvement. It is not adequately detailed in the introduction, with key aspects of the proposed method encapsulated in a single sentence (lines 57-58). The paper would benefit from a more thorough and step-by-step explanation of the modeling method in the early part of the paper, such as the introduction, which would help readers better understand the unique features and operations of the proposed system.

**Questions:**

Missing related references:
- Knowledge Aware Semantic Concept Expansion for Image-Text Matching, IJCAI 2019.

It would be beneficial to include more latest baselines for comparison.


**Limitations:**

The authors adequately addressed the limitations.

---

> ### Author Rebuttal · Authors · 2023-08-09
>
> We thank the reviewer for the feedback and hope to address the questions and concerns below.
> ## Contribution
>
> Though it is an active research area to incorporate commonsense knowledge into AI reasoning, this paper has its unique contribution toward integrating a rule-based symbolic logic reasoner into visual activity understanding, where explicit and rationale rules are encoded to solve real-world complex visual patterns and improve generalization composition.
>
> ## Generality
>
> The method could be more general and facilitate real-world applications. We choose activity understanding as a good and important initial test bed because it is a difficult task with complex visual patterns and compositional nature.
> The proposed symbolic system can be generalized to various tasks (e.g., classification, visual commonsense) with a similar formulation. Some examples are shown below:
>
> 1. Object classification
>     - For the object "crocodile" in CIFAR-100 [1] dataset, one example rule is: long, slightly curved body $\wedge$ four short legs $\wedge$ ... $\wedge$ a long, muscular tail $\rightarrow$ crocodile
>
> 2. Visual commonsense
>     - For one example QA in VCR [2] dataset (shown in attached pdf Fig.1), one example rule is: person on the right shows affectionate gestures $\wedge$ person on the left engages in conversation $\wedge$ ... $\wedge$ The environment is a restaurant $\rightarrow$ b
>
> Under this formulation, semantic coverage and logical entailment remain two important properties under a more general setting, and we can reason with visual inputs via a similar pipeline.
> We conduct experiments with two zero-shot benchmarks:
>
> 1. CIFAR-100 [1] test set.
>
> | Method           | CIFAR-100 Acc (\%) |
> | -----------      |  -----------        |
> | CLIP             | 77.50               |
> | CLIP+Reasoning   | 78.31               |
>
>
> 2. VCR [2] val set. BLIP2 is adopted as a baseline as it is a VQA-style task. VCR val set has 26534 image-question pairs, where we select a subset of 557 pairs for the test. The selected pairs are highly related to human activities (instead of role, mental, .etc) and only involve *one* person's activity.
>
> | Method            | VCR Acc (\%)|
> | -----------       | ----------- |
> | BLIP2             | 46.32        |
> | BLIP2+Reasoning   | 47.08        |
>
> We can see that the proposed pipeline can be applied to general visual tasks and facilitate reasoning. In this paper, we focus on method design instead of task generalization.
> We will add these explanations in the final version and analyze more general settings in future work.
>
> [1] Krizhevsky, Alex, and Geoffrey Hinton. "Learning multiple layers of features from tiny images." (2009): 7.
>
> [2] Zellers, Rowan, et al. "From recognition to cognition: Visual commonsense reasoning." Proceedings of the IEEE/CVF conference on computer vision and pattern recognition. 2019.
>
>
> ## Evaluation Baselines
> We respectfully explain that these more recent vision-language models (LLaVA: Apr 17, MiniGPT-4: Apr 20, and LLaMA Adapter: Mar 28) appeared online within two months of a submission (May 18). According to NeurIPS 2023 policy, they are generally considered "contemporaneous".
> It is beyond our ability to include these vision-language models in submission.
>
> Thanks for your advice to enhance evaluation baselines, which we will add in our final version. We show an additional experiment here, which is based on a more recent vision-language model Otter [1] (May 5) (OTTER-9B-INIT) similar to LLaVA, MiniGPT-4, and LLaMA Adapter. The results are shown below.
>
> | Method      |  Stanford40 zero-shot mAP |
> | ----------- | ----------- |
> | Otter[1]             | 44.83        |
> | Otter[1]+Reasoning   | 47.16        |
>
> Thus, the effectiveness of our method is verified on the SOTA System-1-like method Otter [1].
> Note that, we find the performance of Otter [1] on Stanford40 zero-shot is relatively low compared with CLIP and BLIP2. A possible reason is, the model tuned with visual instructions targets captioning tasks and requires more advanced transitions to solve classification tasks.
> During the time-starved rebuttal period, we try our best to sketch this additional experiment, and a more detailed investigation is left for future work.
>
> [1] Li, Bo, et al. "Otter: A multi-modal model with in-context instruction tuning." arXiv preprint arXiv:2305.03726 (2023).
>
> ## Writing
> Thanks for your advice, we will revise the final version:
>
> - add clearer explanations
> - make sections 3.1-3.2 more concise
> - detail method designs in the early part of the paper
> - add related references

---

> ### Comment · Area_Chair_Gdfa · 2023-08-18
>
> Dear Reviewer sNaH,
>
> We are nearing the end of the discussion period with authors.
>
> The authors have responded in detail to your review, so pls minimally read and acknowledge their rebuttal, and state which (if any) issues you still do not find to be satisfactorily addressed.
>
> You should do so as soon as possible.
>
> Thanks,
> AC

---

### Author Rebuttal · Authors · 2023-08-09

We sincerely thank all reviewers for their constructive comments and recognition of our work’s strengths:

- It is interesting (Reviewer K3QA), encouraging, and worth appreciation (Reviewer zui3) to leverage and extend the intuitive idea of the System-1 and System-2 combination. The contributions are beneficial for the community (Reviewer yAvu).
- The methodology and formalism are reasonable, promising (Reviewer yZAX), well described (Reviewer sNaH, SHNq, yZAX), and easy to follow (Reviewer yZAX). The paper conveys its idea with a good presentation using clear visualizations (Reviewer yAvu).
- Experiments obtain interesting results (Reviewer yAvu) and highlight the contribution (Reviewer SHNq).

We hope to address the questions and concerns via our responses for each review. In the final version, explanations will be added to enhance clarity.
For reading convenience, we summarize some of the major questions raised by reviewers and our response here.

- Generality (Reviewer sNaH, zui3)
    - The proposed symbolic system can be generalized to various tasks (e.g., classification, visual commonsense) with a similar formulation. We choose activity understanding as a good and important initial test bed.
- Computation Scalability (Reviewer SHNq)
    - Symbol predictions will increase the computational cost as a trade-off for explainability and generalization. However, with the proposed method design below, the computational cost can be largely reduced and will not be a major limitation.
- Evaluation Baselines (Reviewer sNaH, SHNq)
    - We explain the baselines of recent VLMs and HOI models.
- Detailed Evaluation\&Analysis (Reviewer K3QA, zui3)
    - We evaluate the proposed two ideal properties, analyze the performance bottleneck and investigate the robustness of the generated ontology.

---

> ### Comment · Area_Chair_Gdfa · 2023-08-13
>
> Dear Reviewers,
>
> The authors have responded in detail to your review feedback, and the current reviewer-author discussion phase ends on Aug 21.
>
> If you have not already done so, pls read the rebuttal(s) and respond within the **next 2-3 days**, so as to allow for further discussion (if needed).
>
> You should read the authors' rebuttal(s) in detail, and ideally:
> 1) Acknowledge the points that you find have been satisfactorily addressed;
> 2) Ask for further clarifications where needed;
> 3) Explain why you may still find certain issues to be insufficiently addressed
>
> Thanks,
> AC

---

> ### Comment · Area_Chair_Gdfa · 2023-08-21
>
> Dear Authors,
>
> On behalf of the reviewers who have not responded (yet), thank you for the detailed responses.
>
> After reading the reviews and rebuttals, I find the clarifications (about symbol selection, recent VLMs, etc.) and further analyses (e.g. bottleneck analysis) to be fairly reasonable. I note the additional results for CIFAR100 and VCR, along with results for Otter. I will give all of these proper consideration.
>
> I do have one general and fairly minor query/comment, which may not be specific to your work alone (but symbolic methods in general). Is it the case that the nature of symbols forces a certain commitment to a fixed, discrete concept (e.g. "hold"), orthogonal to the issues of probabilities and fuzzy logic? More specifically, how can subtleties in the image and/or language modality be handled properly? Using "hand hold onto the side of the boat" as an example, how can/do symbolic methods handle subtleties and variations like "hand hold tightly" vs. "hand hold gently" vs. "hand resting on", which may affect downstream reasoning outcomes?
>
> Thanks,
> AC

---

> > ### Author Response · Authors · 2023-08-21
> >
> > Dear AC,
> >
> > Thanks for your valuable time and efforts and your encouraging feedback. We answer the questions below.
> >
> > The nature of symbols:
> >
> > - Traditional symbolic methods treat one symbol as a fixed, discrete concept in simplified relational tasks (e.g., the BoxWorld problem) where symbols are definitely True/False, e.g., "box1 on box2".
> > - For deep-learning-based visual recognition tasks, symbols are more complex.
> >     - About subtleties: Visual concepts (symbols) are hierarchical. Based on one symbol (e.g., "hand hold"), we can define some finer-grained symbols (e.g., "hand hold tightly") belonging to it.
> >     - About variations: Symbols have different semantic correlations. For example, "hand hold" is more related to "hand rest on" than "hip seated".
> >     - About probabilities and fuzzy logic: The symbols are not definite because there is no pre-defined world model. Thus, the model predicts the symbol probabilities and derives the conclusion via fuzzy logic. Fuzzy logic can be replaced with a gate function applied to probabilities, but it will bring a performance drop due to information loss.
> >
> > How to handle subtleties and variations in our paper:
> >
> > - For the symbolic system construction with an LLM, we use semantic embedding distance as a threshold to judge whether two text descriptions are treated as the same symbol (Alg. 1, L13: redundant symbols removed). Under this standard, "hand hold tightly" and "hand hold gently" are the same symbol as "hand hold", while "hand rest on" and "hand hold" are treated as two separate symbols. In most cases, it avoids activity confusion (detailed in our 1-st response for reviewer yZAX, "Contribution\&Bottleneck Evaluation".2).
> > - For the symbol probabilities of a specific image, the prediction depends on the ability of the adopted System-1-like VLM, e.g., finer-grained verb recognition, and adverb recognition.
> >
> > Possibility of finer-grained symbols:
> >
> > - Better performance may be achieved by a symbolic system with finer-grained symbols, as well as more powerful VLMs to recognize finer-grained symbols.
> > - But it is a performance-cost trade-off. A symbolic system with finer-grained symbols will increase computation costs. Improving a VLM's ability for finer-grained recognition needs more densely annotated data (e.g., adverb dataset [1], visual captioning or VQA datasets, pseudo labels from VLMs, etc.).
> > - We regard it as interesting and important future work to analyze and leverage finer-grained symbols to facilitate reasoning.
> > Thanks again for the insightful feedback!
> >
> > [1] Pang, Bo, et al. "Further understanding videos through adverbs: A new video task." Proceedings of the AAAI Conference on Artificial Intelligence. Vol. 34. No. 07. 2020.
> >
> > Thanks, Authors

---

### Decision · Program_Chairs · 2023-09-21

**Decision:**

Accept (poster)

**Comment:**

Reviewers noted the strengths of this paper to include the important problem being tackled, the novel and promising method, as well as interesting results.

There were initial concerns or questions about: i) comprehensiveness of evaluations, ii) baseline comparison to VLM/LLM without logical system, iii) scalability and iv) understanding the pipeline/system and its bottlenecks. In response, the authors reported an impressive amount of additional work, including results on CIFAR-100, VCR and Stanford-40, a newer VLM model, analysis of System-1 vs System-2 contributions, scalability analysis, etc. The authors also correctly point out that certain SOTA models were released within 2 months of submission date, and hence by NeurIPS policy, these need not be considered.

Overall, the original submission by itself was already a fairly solid piece of work, and the additional results certainly strengthened and generalized the work further (as opposed to patching up flaws). Particularly to encourage non-mainstream approaches, so as to "help the NeurIPS community move faster out of local minima", this work deserves visibility, and I hope that acceptance to NeurIPS will both benefit the community, as well as encourage the authors to take this line of work further.